# Solving Inverse Problems with FLAIR

**Julius Erbach**[1,2]    **Dominik Narnhofer**[1]    **Andreas Dombos**[1]
**Bernt Schiele**[2]    **Jan Eric Lenssen**[2]    **Konrad Schindler**[1]

[1]ETH Zürich        [2]Max Planck Institute for Informatics, Saarland Informatics Campus

## Abstract

Flow-based latent generative models such as Stable Diffusion 3 are able to generate images with remarkable quality, even enabling photorealistic text-to-image generation. Their impressive performance suggests that these models should also constitute powerful priors for inverse imaging problems, but that approach has not yet led to comparable fidelity. There are several key obstacles: (i) the data likelihood term is usually intractable; (ii) learned generative models cannot be directly conditioned on the distorted observations, leading to conflicting objectives between data likelihood and prior; and (iii) the reconstructions can deviate from the observed data. We present FLAIR, a novel, training-free variational framework that leverages flow-based generative models as prior for inverse problems. To that end, we introduce a variational objective for flow matching that is agnostic to the type of degradation, and combine it with deterministic trajectory adjustments to guide the prior towards regions which are more likely under the posterior. To enforce exact consistency with the observed data, we decouple the optimization of the data fidelity and regularization terms. Moreover, we introduce a time-dependent calibration scheme in which the strength of the regularization is modulated according to off-line accuracy estimates. Results on standard imaging benchmarks demonstrate that FLAIR consistently outperforms existing diffusion- and flow-based methods in terms of reconstruction quality and sample diversity. Source code is available at `https://inverseflair.github.io/`.

## 1   Introduction

Flow-based generative models are at the core of modern image generators like Stable Diffusion or FLUX [14]. Beyond image generation based on text prompts, these models have emerged as powerful data-driven priors for a whole range of visual computing tasks. Their comprehensive representation of the visual world, learned from internet-scale training datasets, makes them an attractive alternative to traditional handcrafted image priors. Often, they can be used without any task-specific retraining.

While it is evident that a model capable of generating photorealistic images should be suitable as prior (a.k.a. regularizer) for inverse imaging problems, a practical implementation faces several challenges. On the one hand, flow-based models normally operate in the lower-dimensional latent space of a variational autoencoder (VAE), which means that the forward operator (the relationship between the observed, degraded image and the desired, clean target image) is no longer linear. On the other hand, the iterative nature of the generative process means that intermediate stages are corrupted with (time-dependent) random noise. Hence, one cannot explicitly evaluate their data likelihood, which renders the data term intractable. Moreover, learned generative models tend to overly favor regions of the training distribution that have a high sample density. For test samples that fall in low-density regions, the prior will have a too strong tendency to pull towards outputs with higher a-priori likelihood, compromising fidelity to the input observations.

39th Conference on Neural Information Processing Systems (NeurIPS 2025).

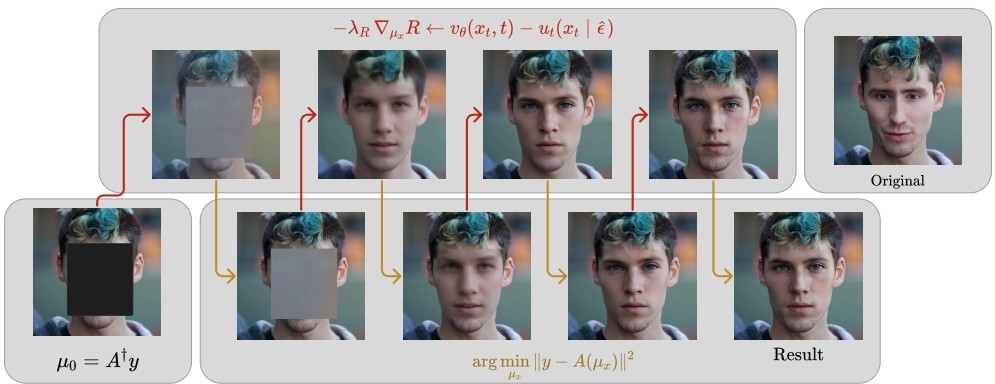

Figure 1: Starting from the adjoint based initialization, we alternate between (i) regularizer updates via a flow-matching loss that aligns the velocity $u_t$ of the variational distribution with the learned velocity field $v_\theta$, and (ii) hard data consistency steps that project the current estimate onto the measurement manifold.

Here, we propose flow-based latent adaptive inference with deterministic re-noising (FLAIR), a novel, training-free variational framework explicitly tailored to integrate flow-based latent diffusion models into inverse problem-solving. To the best of our knowledge, FLAIR is the first scheme that combines latent generative modeling, flow matching and variational inference into a unified formulation for inverse problems. Our main contributions are

- A novel variational objective for inverse problems with flow-matching priors.
- Deterministic trajectory adjustments guide the prior towards regions which are more consistent with the observed data.
- Decoupled optimization of data and regularization terms, enabling hard data consistency.
- A novel, time-dependent weighting scheme, calibrated via offline accuracy estimates, that adapts the regularization along the flow trajectory to match the changing reliability of the model's predictions, ensuring robust inference.

## 2   Related Work

**Deep learning based priors.** Deep learning–based methods typically follow one of two main approaches: they either directly learn an inverse mapping [28, 18, 30, 4, 60], or aim to learn a suitable prior, either through non-generative approaches like unrolled optimization networks [15, 29, 1, 36] or through generative models such as generative adversarial networks [37, 7, 46], or diffusion [17, 49, 52] respectively flow based models [31, 33]. The latter have demonstrated impressive performance in image generation tasks, sparking growing interest in leveraging them as priors for solving inverse problems, particularly through posterior sampling techniques.

**Posterior sampling.**   Although incorporating the implicit prior learned from a denoising-based deep neural network seems straightforward, distinct challenges arise depending on the framework. While early stochastic approaches utilized blind universal denoisers to solve inverse problems [21], they rely on heuristic scheduling and are limited to linear operators. Modern diffusion models formalize this process via fixed forward trajectories, yet this time-dependent structure renders the likelihood intractable [11] A variety of approaches have been proposed for diffusion-based posterior sampling [58, 19, 35, 20]: enforcing the trajectory to stay on the respective noise manifold [11, 12, 61], applying an SVD to run diffusion in the spectral domain [25], utilizing range-null space decomposition during the reverse diffusion process [25], guidance by the pseudo-inverse of the forward operator [50].

Many prior methods perform well in pixel space but are difficult to apply in latent diffusion models due to VAE non-linearity or memory constraints. In order to circumvent this issue, the authors of

ReSample [48] rely on enforcing hard data consistency through optimization and resampling during the reverse diffusion process. PSLD [42], introduces additional objectives terms to ensure that all gradient updates point to the same optima in the latent space. FlowChef [39] incorporates guidance into the flow trajectory during inference, whereas FlowDPS [26] separates the update step into two components: one for estimating the clean image and another for estimating the noise.

In contrast FLAIR follows another class of posterior sampling-based methods, which integrate diffusion priors with inverse problems by directly optimizing a variational objective that approximates the data posterior [34]. This framework was recently extended by RSD [62], which incorporates a repulsion mechanism to promote sample diversity and applied to latent diffusion models. A known issue with this type of optimization is mode collapse [40], which leads to blurry results for these methods. Our method targets this problem by introducing a deterministic trajectory adjustment.

## 3 Background

### 3.1 Inverse problems

In many imaging tasks, such as inpainting [5], super-resolution [38] or tomographic reconstruction [47], one aims to recover a target signal $x \in \mathbb{R}^n$ from a distorted observation $y \in \mathbb{R}^m$. The observation is regarded as the result of applying a forward operator $\mathcal{A} : \mathbb{R}^n \mapsto \mathbb{R}^m$ to the target signal, corrupted by additive Gaussian noise $\nu \in \mathbb{R}^m$ with standard deviation $\sigma_\nu$.

$$y = \mathcal{A}x + \nu. \tag{1}$$

In most practical applications, the forward operator $\mathcal{A}$ is either non-invertible or severely ill-conditioned, making (1) generally ill-posed.

Variational methods solve ill-posed inverse problems by minimizing an energy functional

$$\mathcal{E}(x, y) = \mathcal{D}(x, y) + \mathcal{R}(x). \tag{2}$$

to recover the solution.

Interpreted probabilistically via Bayes' theorem, the posterior distribution $p(x|y)$ is proportional to the product $p(y|x)p(x)$. In the negative log-domain, this yields the data term $\mathcal{D}(x, y) = -\log p(y|x)$ and the regularizer $\mathcal{R}(x) = -\log p(x)$. Handcrafted priors based on regularity assumptions like sparsity [44, 45, 10, 13, 9] have long been the standard, but have largely been replaced by deep learning-based methods in modern data-driven schemes.

### 3.2 Flow based priors

Models based on flow matching [31] learn a time-dependent vector field $v_\theta(x_t, t)$ that continuously transforms samples from a simple initial distribution $p_1(x)$ to a complex target data distribution $p_0(x)$. Formally, this transformation is described by solving the ordinary differential equation (ODE):

$$\frac{\mathrm{d}}{\mathrm{d}t}\psi_t(x) = v_{\theta,t}(\psi_t(x)), \quad t \in [0, 1], \tag{3}$$

where $\psi_t(x)$ represents the trajectory of a sample, evolving smoothly from an initial value drawn at $t = 1$ toward a target value at $t = 0$.

Since the integrated ODE path maps the simple distribution $p_1(x)$ to the complex target $p_0(x)$, the learned flow-based model captures the structure of the data and can therefore serve as a powerful prior for solving inverse problems. To make this approach tractable for high-resolution data, we adopt the latent diffusion model (LDM) framework [41], which shifts the generative process to a lower-dimensional latent space using a pretrained autoencoder with encoder $E : \mathbb{R}^n \mapsto \mathbb{R}^d$ and decoder $D : \mathbb{R}^d \mapsto \mathbb{R}^n$, where $d \ll n$. However, applying such priors to inverse problems introduces challenges, as the non-linearity of the VAE disrupts the linear relationship between measurements and the target signal, resulting in a nonlinear forward operator.

### 3.3 Variational flow sampling

To solve inverse problems from a Bayesian perspective, we aim to sample from the posterior

$$p(x_0|y) \propto p(y|x_0)p(x_0), \tag{4}$$

where the likelihood is given by $p(y|x_0) = \mathcal{N}(\mathcal{A}x_0, \sigma^2\text{Id})$, and $p(x_0)$ represents the prior modeled by the flow-based generative model.

Inspired by previous work [34, 62] we introduce a variational distribution $q(x_0|y) = \mathcal{N}(\mu_x, \sigma_x^2)$ to approximate the true posterior $p(x_0|y)$, by minimizing their Kullback–Leibler divergence:

$$q(x_0|y) \in \arg \min_{q(x_0|y)} \text{KL}(q(x_0|y)\|p(x_0|y)). \tag{5}$$

Rewriting the KL divergence by means of the variational lower bound leads to:

$$KL(q(x_0|y)\|p(x_0|y)) = -\underbrace{\mathbb{E}_{q(x_0|y)}[\log p(y|x_0)]}_{\mathcal{D}(x,y)} + \underbrace{KL(q(x_0|y)\|p(x_0))}_{\mathcal{R}(x)} + \underbrace{\log p(y)}_{\text{const}}. \tag{6}$$

Since a single Gaussian cannot capture a multi-modal posterior, we simplify to a deterministic approximation, setting $\sigma_x^2 = 0$. Equivalently, this corresponds to a single-particle approximation in the sense of Stein variational methods [32]. As shown in [51], rewriting Equation 6 under this approximation and extending it to the time-dependent noisy posterior yields:

$$\arg \min_{q(x_0|y)} \underbrace{\mathbb{E}_{q(x_0|y)}\left[\frac{\|y - f(\mu_x)\|^2}{2\sigma_\nu^2}\right]}_{\mathcal{D}(x,y)} + \underbrace{\int_0^T \omega(t)\, \mathbb{E}_{q(x_t|y)}\left[\|\nabla_x \log q(x_t|y) - \nabla_x \log p(x_t)\|^2\right] dt}_{\mathcal{R}(x)} \tag{7}$$

The first term in Equation 7 describes the data term $\mathcal{D}(x, y)$ and the second the regularizer $\mathcal{R}(x)$, where the integral ensures optimization over the entire diffusion trajectory. Notably, the latter constitutes a weighted score-matching objective, where $\nabla_x \log p(x_t)$ represents the score function [52], which may be extracted from a pretrained diffusion or flow model.

The score of the noisy variational distribution depends on the forward diffusion process and can be computed analytically.

Note that for $\omega(t) = \beta(t)/2$ the weighted score-matching loss recovers the gradient of the diffusion model's evidence lower bound, so that optimizing it yields the maximum likelihood estimate of the data distribution [51]. However, optimizing Equation 7 is costly, as it requires computing the gradient through the flow model. As shown in [54] this can be circumvented by reformulating the regularizer in terms of the Wasserstein gradient flow:

$$\nabla_{\mu_x}\mathcal{R}(x) = \mathbb{E}_{t,q(x_t|y)}\left[\omega(t)(\underbrace{\nabla_x \log q(x_t|y)}_{\text{score of noisy variational distribution}} - \underbrace{\nabla_x \log p(x_t)}_{\text{score of noisy prior distribution}})\right] \tag{8}$$

Note that optimizing only the regularization term, without the data term, at test time is equivalent to the objective of Score Distillation Sampling (SDS) [40].

## 4 Method

**Flow Formulation.** The variational formulation in Equation 7 is formulated for the score, but can be reformulated into a denoising or $\epsilon_\theta$ parameterization [52, 34]. However, we are interested in a variational objective that depends on the velocity field $v_\theta(x_t, t)$, which characterizes the probabilistic trajectory that connects the noise and data distributions.

**Proposition 1.** *We propose to replace the score-based regularizer in the standard variational objective with a flow matching formulation, resulting in the following objective function:*

$$\arg \min_{q(x_0|y)} \underbrace{\mathbb{E}_{q(x_0|y)}\left[\frac{\|y - f(\mu_x)\|^2}{2\sigma_\nu^2}\right]}_{\mathcal{D}(x,y)} + \underbrace{\int_0^T \lambda_\mathcal{R}(t)\, \mathbb{E}_{q(x_t|y)}\left[\|v_\theta(x_t, t) - u_t(x_t|\epsilon)\|^2\right] dt}_{\mathcal{R}(x)} \tag{9}$$

$$\nabla_{\mu_x}\mathcal{R}(x) = \mathbb{E}_{t,q(x_t|y)}\left[\lambda_\mathcal{R}(t)v_\theta(x_t, t) - u_t(x_t \mid \epsilon)\right] \tag{10}$$

The flow-matching term that defines the regularizer arises by reparameterizing the variational distribution to $q(x_t|y) = \mathcal{N}((1-t)\mu_x, t^2 I)$. This corresponds to sampling via the deterministic map $\psi_t(x_0 \mid \epsilon) = (1-t)x_0 + t\epsilon$, with $\epsilon \sim \mathcal{N}(0, I)$. By reformulating the score in terms of the target velocity field $u_t$, we get:

$$\nabla_x \log q(x_t|y) = -\frac{(1-t)u_t(x_t|\epsilon) + x_t}{t} \tag{11}$$

For the learned velocity $v_\theta(x_t, t)$ a similar approximation holds – for a full derivation, see the supplementary material subsection A.3.

$$v_\theta(x_t, t) \approx \frac{-t\nabla_x \log p(x_t) - x_t}{1 - t} \tag{12}$$

We can therefore approximate the score of the noisy prior with our learned velocity field $v_\theta$

$$\nabla_x \log p(x_t) \approx -\frac{(1-t)v_\theta(x_t, t) + x_t}{t}. \tag{13}$$

**Hard Data Consistency.** Existing variational posterior sampling approaches [34, 62] impose soft constraints on the data fidelity term $\mathcal{D}(x, y)$. In contrast, recent work [48] has demonstrated that, when sampling from latent diffusion models, enforcing hard data consistency generally leads to better reconstructions with improved visual fidelity. Our method shares this motivation, but differs in that we optimize over a variational distribution, i.e., we compute $\min \mathbb{E}_{q(x_0|y)}[-\log p(y|x_0)]$. An additional advantage of this variational setup is that it allows us to initialize the optimization variable with an adjoint based initialization $\mu_x = E(A^\top y)$, with $E$ being the encoder of the VAE and $A^\top$ the adjoint of the linear forward operator in pixel space. Other initialization strategies are also possible.

**Accuracy Calibration.** As our framework evaluates the trajectory at each time step, we aim to weight the regularizer's contribution according to its reliability. The difficulty of the prediction task has been shown to depend on the network parameterization, as well as on the specific time step $t$ [22]. Since the regularization term $\mathcal{R}(x)$ in our approach is equivalent to the training objective of the pre-trained flow model, we can easily weight it by the expected model error, which we calibrate on a small set of images. Specifically, we sample $N$ calibration images and compute the conditional flow matching objective for 100 linearly spaced time steps between 0 and 1, then average the error over all images to obtain the expected model error at each time step. Different functions of the model error can be chosen as weight for the regularizer. We choose:

$$\lambda_\mathcal{R}(t) = \frac{1}{N}\left(\sum_{i=1}^{N}\left\|v_\theta(x_t^{(i)}, t) - u_t(x_t^{(i)} \mid \epsilon)\right\|^2\right)^{-1} \tag{14}$$

and set $\lambda_\mathcal{R}(t) = 0$ for all $t < 0.2$, since the accuracy of SD3 is heavily degraded for low noise levels.

**Deterministic Trajectory Adjustment.** Score distillation sampling relies on the assumption that $x_t = (1-t)\mu_x + t\epsilon$ lies in a region of the learned prior that has reasonably high support/density. In practice, this is not always the case. When not tightly conditioned (usually with extensive text prompts), even the best available diffusion models assign low density to many plausible regions of the latent space, leading to bad gradient steps. Therefore, we increase the probability of $p(x_t)$ by additionally conditioning $x_t$ on the estimated "end-point" $\mu$.

**Proposition 2.** *We introduce a reparameterized variational distribution with a mean that linearly interpolates between the posterior mean $\mu_x$ and a model-guided $\hat{x}_1$:*

$$q(x_t \mid y) = \mathcal{N}\left((1-t)\mu_x + t\alpha\hat{x}_1,\ t^2(1-\alpha^2)I\right), \tag{15}$$

*where $\hat{x}_1 = x_{t+\delta t} + (1 - t - \delta t)v_\theta(x_{t+\delta t}, t + \delta t)$ is a single-step velocity-based predictor, and $\alpha \in [0, 1]$ controls the trade-off between deterministic guidance and random noise. This reparameterization induces the following reference velocity field:*

$$u_t(x_t \mid \epsilon) = \frac{\alpha\hat{x}_1 + \sqrt{1-\alpha^2}\epsilon - x_t}{1 - t}. \tag{16}$$

Intuitively, changing the formulation in this manner ensures that the model relocates the sample to its expected position on the learned manifold rather than injecting arbitrary noise, which could drive it in a direction that has high prior likelihood but is not consistent with the observation. To further encourage exploration and avoid collapsing onto the trajectory of the adjoint measurement, we inject an additional stochastic component $\epsilon$ during this process. A full derivation can be found in the supplementary material, subsection A.3.

### 4.1 Algorithm

The following pseudo-code summarizes our method, integrating all the components discussed above.

We adapt the standard scheme [34, 62] of linearly traversing time in a descending manner and stop at $t = 0.2$ as explained in section 4. We choose $\alpha = 1 - t$. Gradient updates to enforce hard data consistency are performed using stochastic gradient descent. For further implementation details and ablations, see subsection A.4

---

**Algorithm 1:** The FLAIR solver for inverse imaging problems

---

**Input:** $\mu_x = \mu_{init}, \lambda_R, \alpha, y, \mathcal{A}, v_\theta$
**Output:** $\mu_x$
$\hat\epsilon \sim \mathcal{N}(0, I)$;          ▷ initial noise sample
**for** $t \leftarrow 1$ **to** $0$ **by** $-\Delta t$ **do**
     $x_t \leftarrow (1 - t)\,\mu_x + t\,\hat\epsilon$;          ▷ sample noisy latent
     $u_t(x_t \mid \hat\epsilon) \leftarrow \dfrac{\hat\epsilon - x_t}{1 - t}$;
     $\nabla_{\mu_x} R \leftarrow v_\theta(x_t, t) - u_t(x_t \mid \hat\epsilon)$;
     $\mu_x \leftarrow \mu_x - \lambda_R \nabla_{\mu_x} R$;          ▷ update w.r.t. regularizer
     $\mu_x \leftarrow \arg\min_{\mu_x} \|y - \mathcal{A}(\mu_x)\|^2$;          ▷ hard data consistency
     $\epsilon \sim \mathcal{N}(0, I)$;
     $\hat{x}_1 \leftarrow x_t + (1 - t)\,v_\theta(x_t, t)$;          ▷ predict deterministic noise
     $\hat\epsilon \leftarrow \alpha\,\hat{x}_1 + \sqrt{1 - \alpha^2}\,\epsilon$;          ▷ update noise estimate

---

## 5 Experiments

We evaluate the performance of FLAIR in a variety of inverse imaging tasks and compare it against several baselines, using the SD3 backbone without any fine-tuning. We used several metrics including SSIM [55], LPIPS [59] and patchwise FID [16] (pFID) to comprehensively assess the perceptual and quantitative quality of the reconstructions. FID is computed using InceptionV3 features on patches of 256x256 resolution. All experiments were performed on a NVidia RTX 4090 GPU with 24GB of VRAM. For completeness we also show PSNR values, but point out that the metric is not well suited for our purposes: PSNR favors the posterior mean, while the goal of the variational approach is to sample from the posterior distribution. Accordingly, PSNR is known to prefer over-smoothed, blurry outputs over sharp ones [6]. To demonstrate that our model can also produce accurate MMSE estimates, we performed ensemble predictions by running posterior sampling eight times and averaging the results. As shown in subsection A.11, ensembling improves PSNR values while reducing LPIPS. This confirms that our samples are distributed around the posterior mean. Moreover, it shows that results closer to the posterior mean – such as those produced by baseline methods – are perceptually farther from the ground truth (in LPIPS) compared to our samples.

### 5.1 Setup

**Datasets.** We utilize two high-resolution image datasets: FFHQ [23] and DIV2K [2]. FFHQ consists of 70k diverse face images at 1024×1024 resolution of which we take the first 1000 samples. It is covering variations in age, pose, lighting, and ethnicity. DIV2K contains 800 high-quality images in 2K resolution that span a range of natural scenes with varied textures and structures.

**Baselines.** Our method is benchmarked against several recent inverse imaging solvers based on posterior sampling. Specifically, we compare to ReSample [48], FlowDPS [26], FlowChef [39], and RSD [62]. The latter is used without repulsive term as it delivers better results. To ensure a fair and meaningful comparison, all methods are evaluated with the same number of function evaluations.

**Problem Setting.** We run and evaluate all methods at a fixed output resolution of $768{\times}768$ pixels. For single image super-resolution, we consider scaling factors of $8\times$ and $12\times$. The corresponding low-resolution inputs are generated by bicubic downsampling. Motion blur is simulated with a blur kernel of size 61. For box inpainting, we mask large, continuous rectangles that cover approximately one third of the observation. All synthesized observations are corrupted with additive Gaussian noise, with standard deviation $\sigma_\nu$ of 0.5%.

For inference on the FFHQ dataset, we use a predefined text prompt of the form *"A high quality photo of a face"*, and for DIV2k *"A high quality photo of"* concatenated with an image-specific description retrieved by applying DAPE [56] to the observation.

## 5.2 Experimental Results

**Inverse Problems.** Our experiments clearly demonstrate that FLAIR outperforms existing flow-based approaches in terms of all perceptual metrics, see Table 1.

In the case of image inpainting, our method produces high-quality reconstructions that fully leverage the power of the generative model and blend naturally into the surrounding context, avoiding degradations and artifacts that we observe in the baselines. In particular, FlowDPS tends to produce implausible textures in the inpainted regions, while FlowChef regularly fails to generate semantically consistent content at all.

For single-image super-resolution, FLAIR consistently delivers the most perceptually convincing and realistic outputs. Notably, the FID scores remain low for both ×8 and ×12 magnification, indicating an effective usage of the generative prior to overcome the increasing ill-posedness. Again, FlowDPS suffers from blur and low texture quality, whereas FlowChef tends to lose semantic coherence.

In motion deblurring, FLAIR also restores sharper and semantically more credible content than competing approaches, which often suffer from residual blur or inconsistent details. The boost in reconstruction quality is quantitatively reflected by all metrics, confirming that FLAIR reconstructs images with high fidelity. For further qualitative examples, see subsection A.13.

Table 1: **Quantitative results** with 50 NFE and $\sigma_\nu = 0.5\%$.

| Method | SR ×8 | | | | SR ×12 | | | | Motion Deblurring | | | | Inpainting | | | |
|---|---|---|---|---|---|---|---|---|---|---|---|---|---|---|---|---|
| | LPIPS↓ | FID↓ | SSIM↑ | PSNR↑ | LPIPS↓ | FID↓ | SSIM↑ | PSNR↑ | LPIPS↓ | FID↓ | SSIM↑ | PSNR↑ | LPIPS↓ | FID↓ | SSIM↑ | PSNR↑ |
| **FFHQ 768×768** | | | | | | | | | | | | | | | | |
| ReSample | 0.400 | 55.6 | **0.815** | 26.37 | 0.474 | 80.3 | **0.786** | 25.47 | 0.457 | 82.9 | **0.788** | 25.45 | 0.366 | 70.8 | 0.827 | 21.83 |
| FlowDPS | 0.374 | 38.5 | 0.756 | 29.24 | 0.413 | 44.0 | 0.741 | 28.05 | 0.431 | 54.3 | 0.732 | 27.64 | 0.344 | 42.5 | 0.771 | 19.19 |
| RSD | 0.391 | 51.7 | 0.776 | **29.69** | 0.462 | 71.7 | 0.743 | **28.11** | 0.458 | 77.3 | 0.743 | 27.67 | 0.478 | 73.3 | 0.736 | 21.97 |
| FlowChef | 0.341 | 30.5 | 0.760 | 28.42 | 0.373 | 46.5 | 0.730 | 27.00 | 0.406 | 40.2 | 0.716 | 25.81 | 0.394 | 69.8 | 0.780 | 18.18 |
| **Ours** | **0.213** | **13.3** | 0.777 | 29.54 | **0.271** | **16.2** | 0.740 | 27.71 | **0.236** | **10.7** | 0.772 | **29.61** | **0.184** | **8.7** | **0.828** | **23.69** |
| **DIV2K 768×768** | | | | | | | | | | | | | | | | |
| ReSample | 0.533 | 55.0 | **0.625** | 22.34 | 0.643 | 88.1 | **0.562** | 20.85 | 0.556 | 79.7 | 0.617 | 21.79 | 0.285 | 51.9 | 0.796 | 22.68 |
| FlowDPS | 0.476 | 44.4 | 0.567 | 23.01 | 0.547 | 54.0 | 0.528 | 21.79 | 0.558 | 65.5 | 0.536 | 21.88 | 0.328 | 29.2 | 0.692 | 21.71 |
| RSD | 0.539 | 60.9 | 0.591 | **23.45** | 0.684 | 95.7 | 0.523 | **21.96** | 0.638 | 97.6 | 0.551 | 22.10 | 0.464 | 63.9 | 0.678 | 23.23 |
| FlowChef | 0.490 | 36.5 | 0.539 | 21.84 | 0.525 | 43.8 | 0.492 | 20.52 | 0.561 | 49.6 | 0.486 | 19.90 | 0.489 | 58.3 | 0.659 | 20.87 |
| **Ours** | **0.353** | **26.5** | 0.607 | 23.30 | **0.421** | **32.1** | 0.525 | 21.39 | **0.315** | **21.1** | **0.653** | **24.44** | **0.163** | **11.0** | **0.815** | **23.75** |

**Posterior Variance.** To demonstrate that FLAIR does not suffer from mode collapse, we assess the posterior variance $\text{Var}[x|y]$ for the task of ×12 Super Resolution, by drawing 32 samples for a fixed observation $y$ and computing their pixel-wise variance. We conduct that experiment for our FLAIR approach, for RSD with repulsive term, and for FlowDPS [26]. The example in Figure 3 illustrates that FLAIR has the highest sample diversity, which is also reflected in the corresponding variance maps. Notably, the sample variance is concentrated in regions with high-frequency textures. This indicates that our method reliably reconstructs the posterior, whose low-frequency part is, in the super-resolution setting, tightly constrained by the likelihood term.

**Editing.** Beyond image restoration, we observe that our method also performs remarkably well for text-based image editing, simply presenting suitable target prompts during inpainting. Figure 4 illustrates a variety of edited images generated from the same photograph with the help of the depicted masks and prompts.

**Pixel Space Experiments** We also implement FLAIR in pixel-space using the model from [33], trained on CelebA-HQ resized to 256x256 px. We compare to DDNM [53], DPS [11], Moment Matching [43] and ΠGDM [50]. We tuned the hyperparameters for all baselines, which we report in subsection A.9. As shown in Table 5.2, our method also outperforms previous work in the pixel space, demonstrating its broader applicability.

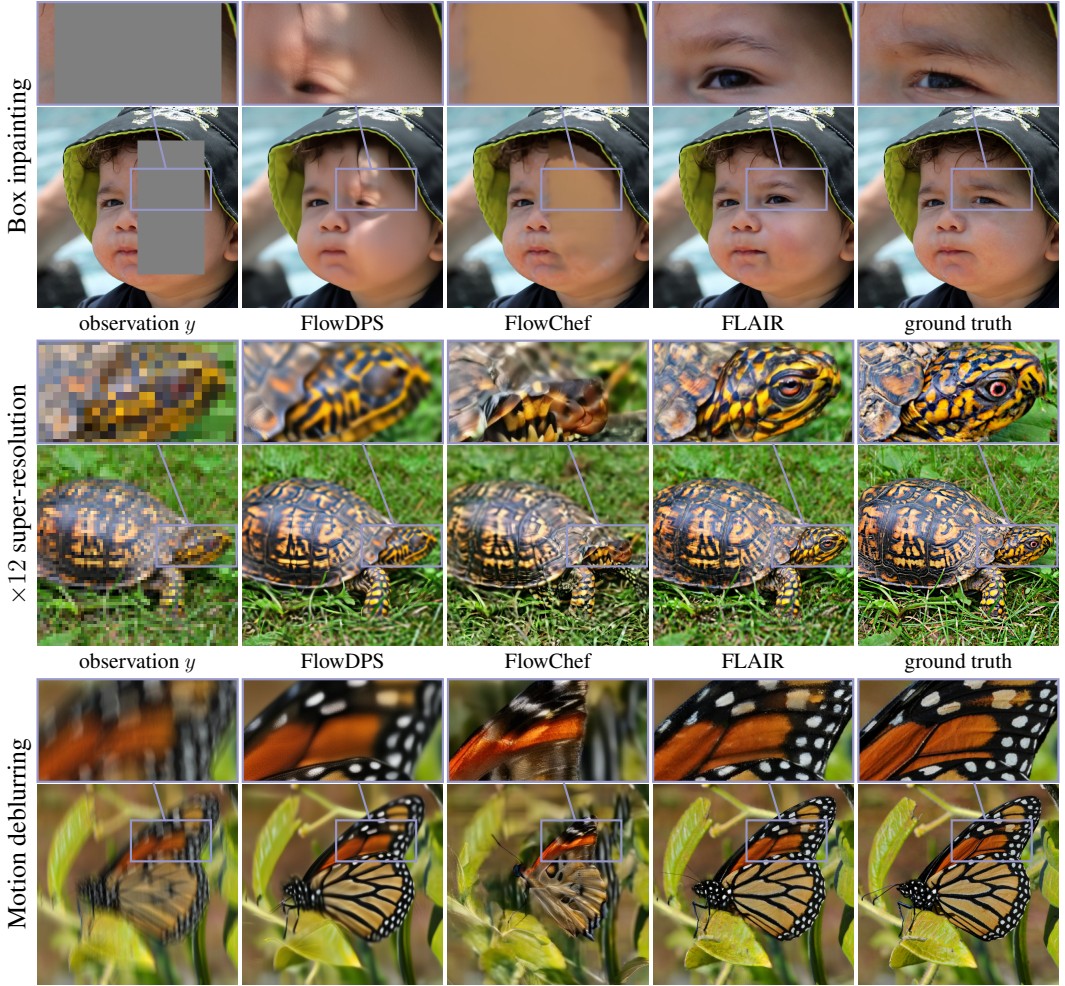

Figure 2: **Qualitative comparison**. FLAIR produces posterior samples of high perceptual quality while maintaining high data likelihood. Best viewed zoomed in.

Table 2: **Quantitative results** with 50 NFE and $\sigma_\nu = 0.5\%$ – In-painting and Super-resolution ($\times 8$).

| | Inpainting | | | | SR $\times 8$ | | | |
|---|---|---|---|---|---|---|---|---|
| Method | LPIPS↓ | FID↓ | SSIM↑ | PSNR↑ | LPIPS↓ | FID↓ | SSIM↑ | PSNR↑ |
| DDNM | $\underline{0.158 \pm 0.042}$ | $\underline{26.9}$ | $\underline{0.732 \pm 0.037}$ | $18.31 \pm 2.94$ | $0.199 \pm 0.052$ | $31.9$ | $0.635 \pm 0.079$ | $23.59 \pm 1.64$ |
| DPS | $0.195 \pm 0.064$ | $30.2$ | $0.689 \pm 0.077$ | $20.49 \pm 2.81$ | $0.172 \pm 0.058$ | $27.8$ | $0.658 \pm 0.088$ | $24.59 \pm 2.04$ |
| MM | $0.161 \pm 0.054$ | $28.8$ | $0.728 \pm 0.062$ | $\underline{20.59 \pm 3.37}$ | $0.172 \pm 0.051$ | $29.1$ | $0.669 \pm 0.083$ | $24.65 \pm 1.97$ |
| ΠGDM | $0.195 \pm 0.064$ | $30.2$ | $0.689 \pm 0.077$ | $20.49 \pm 2.81$ | $\underline{0.157 \pm 0.052}$ | $\underline{26.5}$ | $0.677 \pm 0.084$ | $\underline{24.98 \pm 2.07}$ |
| **FLAIR** | $\mathbf{0.097 \pm 0.035}$ | $\mathbf{14.2}$ | $\mathbf{0.831 \pm 0.031}$ | $\mathbf{21.87 \pm 2.66}$ | $\mathbf{0.143 \pm 0.039}$ | $\mathbf{22.9}$ | $\mathbf{0.712 \pm 0.076}$ | $\mathbf{25.93 \pm 1.96}$ |

## 5.3 Ablation Studies

We systematically analyze the impact of key design choices in our method. Specifically, we ablate the deterministic trajectory adjustment, the use of hard data consistency, and the calibration of the regularizer weight for ×12 super-resolution, using a subset of 100 samples from the FFHQ and DIV2K datasets. Quantitative and qualitative results are shown in Table 3 and Figure 6, respectively.

**Hard Data Consistency (HDC).** Dropping the hard data consistency degrades both metrics, with PSNR being particularly affected due to poorer alignment with the input observation, which is also evident in the visual example: the reconstruction is plausible but deviates from the observation.

**Deterministic Trajectory Adjustment (DTA).** The biggest performance drop compared to the full setup occurs when removing the deterministic trajectory adjustment, as random noise sampling harms

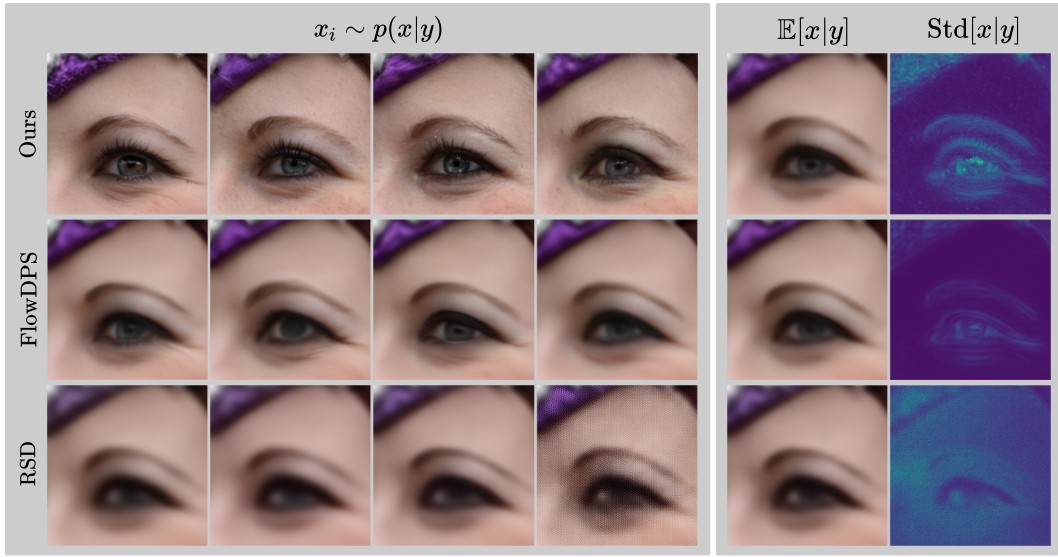

Figure 3: **Zoomed-in reconstructions for x12 Super Resolution.** We show posterior samples (col. 1–4) of FLAIR, FlowDPS, and RSD, posterior mean and standard deviation (over 32 samples, col. 5,6). 0 ▬▬▬ 0.16

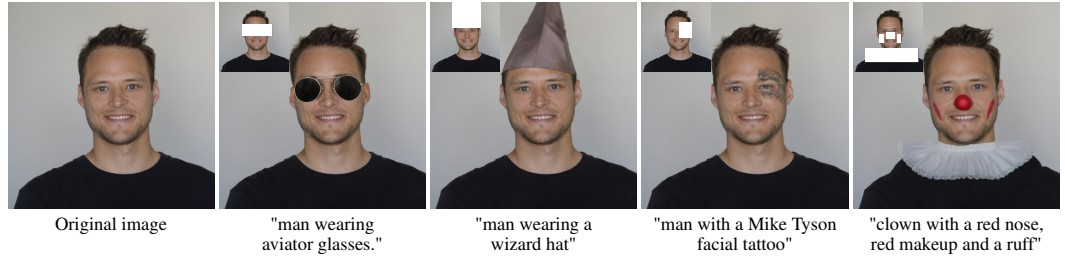

| Original image | "man wearing aviator glasses." | "man wearing a wizard hat" | "man with a Mike Tyson facial tattoo" | "clown with a red nose, red makeup and a ruff" |

Figure 4: **Edited images** shown alongside original, with prompts: "A high resolution portrait of a..."

the gradient updates in low-density regions of the prior. The reconstruction appears overly smooth and lacks texture details.

**Calibrated Regularizer Weight (CRW).** Replacing our calibrated regularizer weight with $\lambda_{\mathcal{R}}(t) = t$ also has a strong impact on perceptual quality: the result is visibly blurred if one ignores the changing accuracy of the regularizer along the flow trajectory.

Table 3: **Ablation study** for $\times 12$ super-resolution on DIV2K and FFHQ. Model components are individually switched on or off.

| HDC | DTA | CRW | FFHQ | | DIV2K | |
|-----|-----|-----|------|------|-------|------|
| | | | LPIPS ↓ | PSNR ↑ | LPIPS ↓ | PSNR ↑ |
| ✓ | ✓ | ✓ | 0.259 | 27.45 | 0.427 | 21.05 |
| ✗ | ✓ | ✓ | 0.297 | 27.17 | 0.467 | 20.82 |
| ✓ | ✗ | ✓ | 0.432 | 27.20 | 0.622 | 21.69 |
| ✓ | ✓ | ✗ | 0.363 | 28.58 | 0.583 | 21.98 |
| ✗ | ✗ | ✗ | 0.392 | 28.33 | 0.605 | 21.99 |

**Legend.** HDC: Hard Data Consistency; DTA: Deterministic Trajectory Adjustment; CRW: Calibrated Regularizer Weight. ✓ = included, ✗ = ablated.

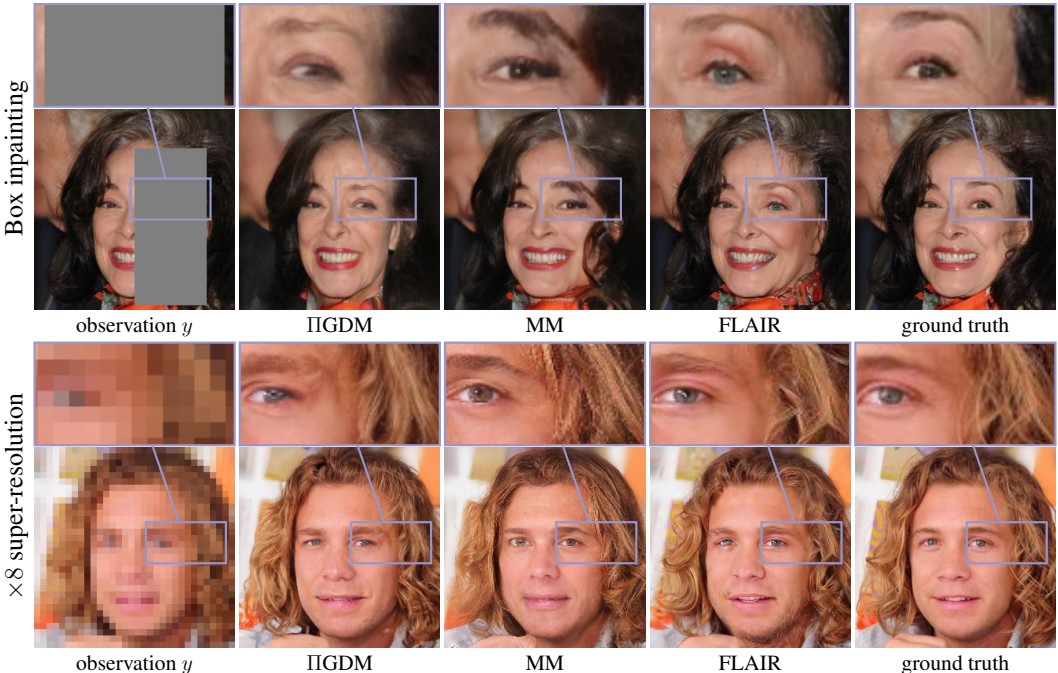

Figure 5: **Qualitative comparison.** FLAIR in pixel space produces posterior samples.

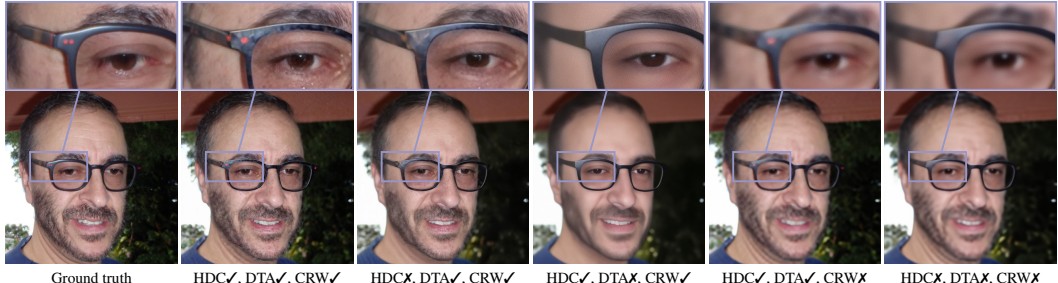

Figure 6: **Qualitative samples from the ablation study** on ×12 Super Resolution.

## 6  Conclusion and Limitations

We have presented FLAIR, a training-free variational framework for inverse problems that uses a flow-based generative model as its image prior. By combining the power of (latent) flow-based models with a principled reconstruction of the posterior distribution, FLAIR addresses key limitations of existing methods. First, it is able to target the generation towards images, which match the observation, by aiding the degradation-agnostic flow matching loss with deterministic noise vectors. Second, it enables hard data consistency without sacrificing sample diversity, by decoupling the data consistency constraint from the regularization, while adaptively reweighting the latter according to its expected accuracy, calibrated offline. Experiments with different image datasets and tasks confirm that FLAIR consistently achieves higher reconstruction quality than existing baselines based on either flow matching or denoising diffusion. Notably, our proposed method achieves, at the same time, excellent perceptual quality, close adherence to the input observations, and high sample diversity.

Evidently, FLAIR inherits the limitations of the underlying generative model. These include biases caused by the selection of training data, constraints w.r.t. the output resolution, and a limited ability to recover out-of-distribution modes. Furthermore, our approach introduces additional hyper-parameters needed to control the deterministic trajectory adjustment. We note that high fidelity image restoration methods can potentially be misused for unethical image manipulations.

**Acknowledgments** This work was funded, in part, by the Max Plank ETH Center for Learning Systems and Huawei Technologies Oy (Finland) Co. Ltd.

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

# Supplementary Material

In the following, we provide detailed line-by-line derivations of the mathematical formulations used in the paper, as well as additional implementation details and experimental results.

## A   Derivations

### A.1   Derivation of flow-based variational formulation

The linear conditional flow and it's corresponding velocity are defined as:

$$\psi_t(x_0 \mid \epsilon) = (1-t)\,x_0 + t\,\epsilon, \quad \epsilon \sim \mathcal{N}(0, I) , \tag{17}$$

$$u_t(x_t|\epsilon) = \frac{\mathrm{d}\psi_t}{\mathrm{d}t}(\psi_t^{-1}(x_t|\epsilon)|\epsilon) . \tag{18}$$

The score of the noisy variational distribution can be analytically computed with:

$$q(x_t|y) = \mathcal{N}((1-t)\mu_x, t^2\, I) , \tag{19}$$

$$\nabla_{x_t} \log q(x_t|y) = -\frac{\epsilon}{t} . \tag{20}$$

We compute $\frac{\mathrm{d}\psi_t}{\mathrm{d}t}(x_0|\epsilon) = -x_0 + \epsilon$ and $\psi_t^{-1}(x_t|\epsilon)$ and insert it into Equation 18:

$$u_t(x_t|\epsilon) = \frac{\epsilon - x_t}{1 - t} . \tag{21}$$

Solving Equation 21 for $\epsilon$ and inserting in Equation 20 gives:

$$\nabla_{x_t} \log q(x_t|y) = -\frac{(1-t)u_t(x_t|\epsilon) + x_t}{t} . \tag{22}$$

For the learned velocity $v_\theta(x_t, t)$ a similar approximation holds:

$$v_\theta(x_t, t) \approx \frac{-t\nabla_x \log p(x_t) - x_t}{1 - t} . \tag{23}$$

Hence, we can approximate the score of the noisy prior with our learned velocity field $v_\theta$

$$\nabla_{x_t} \log p(x_t) \approx -\frac{(1-t)v_\theta(x_t, t) + x_t}{t} , \tag{24}$$

and we see that for $\omega(t) = \frac{t}{1-t}$ we obtain the conditional flow matching objective for $\mathcal{R}(x)$. We therefore set $\omega(t) = \frac{t}{1-t}$ and end up at our final objective:

$$\arg\min_{q(x_0|y)} \mathbb{E}_{q(x_0|y)} \underbrace{\left[ \frac{\|y - f(\mu_x)\|^2}{2\nu^2} \right]}_{\mathcal{D}(x, y)} + \underbrace{\int_0^T \mathbb{E}_{q(x_t|y)} \left[ \|v_\theta(x_t, t) - u_t(x_t|\epsilon)\|^2 \right] dt}_{\mathcal{R}(x)} . \tag{25}$$

Again, the gradient step for the regularizer becomes:

$$\nabla_{\mu_x} \mathcal{R}(x) = \mathbb{E}_{t, q(x_t|y)} \left[ v_\theta(x_t, t) - u_t(x_t|\epsilon) \right] . \tag{26}$$

### A.2   Derivation of trajectory adjusted flow-based variational formulation

To achieve the proposed trajectory adjustment, we modify the forward process to:

$$\hat{x}_1 = x_{t+dt} + (1 - t - dt)v_\theta(x_{t+dt}, t + dt) , \tag{27}$$

$$x_t = (1-t)\mu_x + t\underbrace{\left(\alpha\hat{x}_1 + \sqrt{1 - \alpha^2}\epsilon\right)}_{\hat{\epsilon}} , \tag{28}$$

where $\hat{x}_1$ is the noise vector prediction from the last optimization iteration. This induces a variational distribution:

$$q(x_t \mid y) = \mathcal{N}\left((1-t)\mu_x + t\alpha\hat{x}_1, \, t^2(1-\alpha^2)I\right) , \tag{29}$$

leading to a score of

$$\nabla_{x_t} \log q(x_t \mid y) = -\frac{1}{t^2(1-\alpha^2)} \cdot t\sqrt{1-\alpha^2}\epsilon = -\frac{\epsilon}{t\sqrt{1-\alpha^2}} . \tag{30}$$

The velocity field is again computed by Equation 18. We start by defining the flow:

$$\psi_t(x_0 \mid \epsilon) = (1-t)x_0 + t\left(\alpha\hat{x}_1 + \sqrt{1-\alpha^2}\epsilon\right) . \tag{31}$$

The resulting derivative reads

$$\frac{d}{dt}\psi_t(x_0 \mid \epsilon) = \alpha\hat{x}_1 - x_0 + \sqrt{1-\alpha^2}\epsilon , \tag{32}$$

and the inverse becomes

$$x_0 = \psi_t^{-1}(x_t \mid \epsilon) = \frac{x_t - t\alpha\hat{x}_1 - t\sqrt{1-\alpha^2}\epsilon}{1-t} . \tag{33}$$

Plugging these results into Equation 18:

$$u_t(x_t \mid \epsilon) = \frac{\alpha\hat{x}_1 + \sqrt{1-\alpha^2}\epsilon - x_t}{1-t} . \tag{34}$$

## A.3 Derivation of Score from Flow

The score matching objective reads as:

$$\nabla_{x_t} \ln p_t(x_t) = \arg\min_{\theta} \mathbb{E}_{t\sim\mathcal{U}[0,1],x_0\sim p_0,\epsilon\sim\mathcal{N}(0,I)} \left[ w(t) \cdot \left\| s_\theta(x_t,t) + \frac{1}{\sigma(t)^2}\left(x_t - \mu(x_0,t)\right) \right\|^2 \right], \tag{35}$$

where,

$$-\frac{1}{\sigma(t)^2}\left(x_t - \mu(x_0,t)\right) = \nabla_{x_t} \log p_t(x_t \mid x_0), \tag{36}$$

with $p_t(x_t \mid x_0) = \mathcal{N}(\mu_t(x_0), \sigma_t^2 I)$. Note that as usual we assume $\mu_t(x_0)$ being linear in $x_0$. Equation 35 is solved by:

$$\nabla_{x_t} \log p_t(x_t) = \mathbb{E}_{p_t(x_0|x_t)}\left[\nabla_{x_t} \log p_t(x_t|x_0)\right], \tag{37}$$

and can be written as:

$$\nabla_{x_t} \log p_t(x_t) = \frac{-(x_t - \mu(\mathbb{E}[x_0 \mid x_t], t))}{\sigma(t)^2}. \tag{38}$$

In the case of OT flow-matching, we obtain

$$x_t = (1-t)x_0 + tx_1, \tag{39}$$

$x_1 \sim \mathcal{N}(0, \mathrm{Id})$ and $p(x_t|x_0) = \mathcal{N}((1-t)x_0, t^2)$. The optimal velocity under the flow matching loss is given by:

$$v^*(x_t, t) = \mathbb{E}[x_1 - x_0 \mid x_t]. \tag{40}$$

Expressing $x_1 = \frac{x_t - (1-t)x_0}{t}$, we can insert into Equation 40 and obtain:

$$\mathbb{E}[x_0 \mid x_t] = x_t - t\mathbb{E}[x_1 - x_0 \mid x_t]. \tag{41}$$

Inserting in Equation 38 leads to:

$$\nabla_{x_t} \log p_t(x_t) = -\frac{x_t - (1-t)(x_t - t\mathbb{E}[x_1 - x_0 \mid x_t])}{t^2}, \tag{42}$$

which for $v^*(x_t, t) = \mathbb{E}[x_1 - x_0 \mid x_t]$ reads as:

$$\nabla_{x_t} \log p(x_t) \approx -\frac{(1-t)v_\theta(x_t, t) + x_t}{t}. \tag{43}$$

### A.4 Implementation details

**Flow Model and Regularizer Settings.** As flow matching model, we us Stable Diffusion 3.5-Medium, which has been released under the Stability Community License. The classifier-free guidance scale is set to 2 for all experiments. To minimize the regularization term, we use stochastic gradient descent with a learning rate of 1.

**Data Likelihood Term.** We use stochastic gradient descent for the minimization of the data term towards hard data consistency. For numerical stability, the squared error is summed over all measurements instead of computing the mean. The learning rate has to be adjusted accordingly, to compensate for the varying number of measurements $y$. Moreover, the minimization is terminated with early stopping once the likelihood term reaches $1 \times 10^{-4} \cdot \text{len}(y)$, to not overfit the noise in the image observation.

**Super-resolution.** We employ bicubic downsampling as the forward operator, as implemented in [53]. The learning rate is set to 12 for $\times 12$ super-resolution and to 6 for $\times 8$ super-resolution.

**Motion Deblurring.** A different motion blur kernel is created for each sample using the *MotionBlur* package [8], available via github, with kernel size 61 and intensity 0.5. The learning rate for our data term optimizer is set to $10^{-1}$.

**Inpainting.** For inpainting on FFHQ we always use the same rectangular mask at a fixed position, chosen such that it roughly masks out the right side of the face (Figure 3). For DIV2k we also use a fixed mask for all samples, consisting of six randomly generated rectangles (Figure 6).

**Data.** We use the publicly available Flickr Faces High Quality dataset [24], which is realeased under the Creative Commons BY 2.0 License and the DIV2K dataset [3], which is released under a research only license. For FFHQ we use the first 1000 samples of the evaluation dataset and for DIV2K we use the 800 training samples. We downscale both datasets to $768 \times 768$ px by applying bicubic sampling so that the shorter edge of the frame has 768 px and apply central cropping afterwards.

### A.5 Baselines

For comparability, all baselines use Stable Diffusion 3.5-Medium and the same task definitions as in A.4.

**FlowDPS** [26] The standard FlowDPS implementation [27] is applied with 50 NFE, a classifier-free guidance scale of 2, and step sizes of 15 for inpainting and 10 for all other tasks.

**FlowChef** [39] Additionally, [27] is employed for FlowChef as well, using 200 NFE for inpainting and 50 NFE for all other tasks, a classifier-free guidance scale of 2, and a step size of 1 for all tasks.

**Repulsive Score Distillation (RSD)** [62]. We implement RSD for flow-matching models by applying Proposition 1 with $\omega(t) = t$, resulting in a weighting term consistent with the original RSD approach. However, we omit the pixel-space augmentation as it negatively affected performance when combined with the SD3 VAE. Consistent with the original findings from RSD, we observed that incorporating the repulsive term improves sample diversity but reduces fidelity. Therefore, we set the repulsive term to 0 for all results presented in the table, employing it exclusively for comparing posterior variances.

**ReSample** [48] We re implement ReSample for flow-matching by setting $\bar{\alpha}_t = \frac{(1-t)^2}{t^2 + (1-t)^2}$. Furthermore, we compute the hard-data consistency at every iteration as larger skip steps seem to harm performance. We set the learning rate of the data term optimizer to 15 for all inverse problems.

**PSLD** [42] Our attempt to adapt PSLD following [27]—using 500 NFE, a classifier-free guidance scale of 2, and step sizes of 1 ($\times 12$ super-resolution), 0.5 ($\times 8$ super-resolution and motion deblurring) and 0.1 (inpainting)—did not yield meaningful results.

### A.6 Regularizer weighting

Figure 1 displays the mean and standard deviation of the conditional flow matching loss $\mathcal{L}_{CFM}$ as a function of $t$, estimated over 100 samples. The loss function starts with high values at $t = 1$, decreases over time, but then starts to rise again, and when reaching $t \approx 0.2$ even exceeds its initial value . The rising loss when approaching $t = 0$ is due, in part, to the increasing difficulty of distinguishing high-frequency image content from residual noise. Another factor is that near $t = 0$ the model

operates in a highly sensitive regime where small prediction errors can cause disproportionately large deviations from the target, making accurate flow estimation particularly challenging in the final stages of the trajectory. We therefore modulate the regularization term according to the model error. Different weighting functions for $f(\mathcal{L}_{CFM})$ could be chosen that fulfill the condition $\lambda_{\mathcal{R}(t=0)} = 0$. We simply take the reciprocal of the model error $\lambda_{\mathcal{R}(t)} = \mathcal{L}_{CFM,t}^{-1}$ as the regularization weight while $t \geq 0.2$, then set it to 0 for $t < 0.2$. An alternative would be to shift the reciprocal of $\mathcal{L}_{CFM,t}^{-1}$ by $\mathcal{L}_{CFM,t=0}^{-1}$, such that $\lambda_{\mathcal{R}(t)} = \mathcal{L}_{CFM,t}^{-1} - \mathcal{L}_{CFM,t=0}^{-1}$. In Table 1 we compare our default weighting with this variant, denoted as $\lambda_{shift}$.

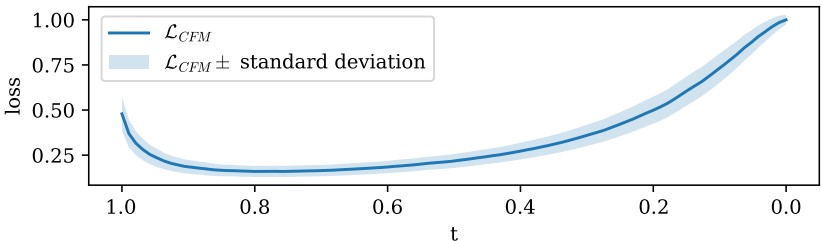

Figure 1: The Flow-Matching loss over time $t$.

Table 1: **Quantitative results** with 50 NFE and $\sigma_\nu = 0.01$. We compare different weighting functions $\lambda_{\mathcal{R}(t)}$ based on the model error

| | SR ×8 | | | | SR ×12 | | | | Motion Deblurring | | | | Inpainting | | | |
|---|---|---|---|---|---|---|---|---|---|---|---|---|---|---|---|---|
| Method | LPIPS↓ | FID↓ | SSIM↑ | PSNR↑ | LPIPS↓ | FID↓ | SSIM↑ | PSNR↑ | LPIPS↓ | FID↓ | SSIM↑ | PSNR↑ | LPIPS↓ | FID↓ | SSIM↑ | PSNR↑ |
| **FFHQ 768×768** | | | | | | | | | | | | | | | | |
| $\lambda_{shift}$ | 0.246 | 27.4 | **0.793** | **29.91** | 0.286 | 24.4 | **0.766** | **28.19** | 0.237 | 14.5 | **0.790** | **29.84** | **0.180** | **8.2** | **0.828** | 23.58 |
| **Ours** | **0.213** | **13.3** | 0.777 | 29.54 | **0.271** | **16.2** | 0.740 | 27.71 | **0.236** | **10.7** | 0.772 | 29.61 | 0.184 | 8.7 | 0.828 | **23.69** |
| **DIV2K 768×768** | | | | | | | | | | | | | | | | |
| $\lambda_{shift}$ | 0.379 | 30.4 | **0.625** | **23.58** | 0.434 | 37.5 | 0.522 | **21.40** | 0.337 | 25.8 | **0.664** | **24.54** | **0.151** | **9.0** | **0.819** | **23.79** |
| **Ours** | **0.353** | **26.5** | 0.607 | 23.30 | **0.421** | **32.1** | **0.525** | 21.39 | **0.315** | **21.1** | 0.653 | 24.44 | 0.163 | 11.0 | 0.815 | 23.75 |

## A.7 Effect of captioning

Given the diversity of DIV2k, we use DAPE [57] to generate captions for it and include them in the prompt *A high quality photo of [DAPE caption].* For FFHQ we always prompt with *A high quality photo of a face.*. The effect of the text prompt is to increase the likelihood of our sample under the prior of the (pre-trained, frozen) image generator. For comparison, we also ran experiments without data specific captions, where we always used the generic prompt *A high quality photo.* Results are shown in Table 2

Table 2: Quantitative results with 50 NFE and $\sigma_\nu = 0.01$. We compare our version with data-specific captions and a version without captions.

| | FFHQ | | | | DIV2K | | | |
|---|---|---|---|---|---|---|---|---|
| Method | LPIPS↓ | FID↓ | SSIM↑ | PSNR↑ | LPIPS↓ | FID↓ | SSIM↑ | PSNR↑ |
| wo captions | 0.278 | 17.0 | 0.734 | 27.66 | 0.488 | 51.5 | **0.546** | **21.82** |
| **Ours** | **0.271** | **16.2** | **0.740** | **27.71** | **0.421** | **32.1** | 0.525 | 21.39 |

## A.8 Additional Experimental Results

We present the experimental results from the main paper in Table 3, now augmented with sample-wise standard deviations for all metrics except FID.

Table 3: **Quantitative results** with 50 NFE and $\sigma_\nu = 0.01$ – Super-resolution ($\times 8$ and $\times 12$).

| | SR $\times 8$ | | | | SR $\times 12$ | | | |
|---|---|---|---|---|---|---|---|---|
| Method | LPIPS↓ | FID↓ | SSIM↑ | PSNR↑ | LPIPS↓ | FID↓ | SSIM↑ | PSNR↑ |
| **FFHQ 768×768** | | | | | | | | |
| ReSample | $0.400 \pm 0.069$ | 55.6 | $\mathbf{0.815 \pm 0.051}$ | $26.37 \pm 1.00$ | $0.474 \pm 0.078$ | 80.3 | $\mathbf{0.786 \pm 0.056}$ | $25.47 \pm 1.16$ |
| FlowDPS | $0.374 \pm 0.107$ | 38.5 | $0.756 \pm 0.075$ | $29.24 \pm 2.04$ | $0.413 \pm 0.107$ | 44.0 | $0.741 \pm 0.074$ | $28.05 \pm 2.06$ |
| RSD | $0.391 \pm 0.079$ | 51.7 | $0.776 \pm 0.052$ | $\mathbf{29.69 \pm 2.04}$ | $0.462 \pm 0.093$ | 71.7 | $0.743 \pm 0.059$ | $\mathbf{28.11 \pm 2.00}$ |
| FlowChef | $0.341 \pm 0.083$ | 30.5 | $0.760 \pm 0.064$ | $28.42 \pm 2.22$ | $0.373 \pm 0.084$ | 46.5 | $0.730 \pm 0.068$ | $27.00 \pm 2.07$ |
| **Ours** | $\mathbf{0.213 \pm 0.056}$ | **13.3** | $0.777 \pm 0.051$ | $29.54 \pm 2.02$ | $\mathbf{0.271 \pm 0.071}$ | **16.2** | $0.740 \pm 0.058$ | $27.71 \pm 2.00$ |
| **DIV2K 768×768** | | | | | | | | |
| ReSample | $0.533 \pm 0.130$ | 55.0 | $\mathbf{0.625 \pm 0.132}$ | $22.34 \pm 2.27$ | $0.643 \pm 0.152$ | 88.1 | $\mathbf{0.562 \pm 0.151}$ | $20.85 \pm 3.02$ |
| FlowDPS | $0.476 \pm 0.129$ | 44.4 | $0.567 \pm 0.139$ | $23.01 \pm 3.01$ | $0.547 \pm 0.139$ | 54.0 | $0.528 \pm 0.146$ | $21.79 \pm 2.94$ |
| RSD | $0.539 \pm 0.121$ | 60.9 | $0.591 \pm 0.124$ | $\mathbf{23.45 \pm 2.96}$ | $\mathbf{0.684 \pm 0.137}$ | 95.7 | $0.523 \pm 0.132$ | $\mathbf{21.96 \pm 2.86}$ |
| FlowChef | $0.490 \pm 0.116$ | 36.5 | $0.539 \pm 0.137$ | $21.84 \pm 2.96$ | $0.525 \pm 0.118$ | 43.8 | $0.492 \pm 0.145$ | $20.52 \pm 2.85$ |
| **Ours** | $\mathbf{0.353 \pm 0.112}$ | 26.5 | $0.607 \pm 0.127$ | $23.30 \pm 2.90$ | $\mathbf{0.421 \pm 0.131}$ | **32.1** | $0.525 \pm 0.136$ | $21.39 \pm 2.67$ |

Table 4: **Quantitative results** with 50 NFE and $\sigma_\nu = 0.01$ – Motion deblurring and in-painting.

| | Motion Deblurring | | | | In-painting | | | |
|---|---|---|---|---|---|---|---|---|
| Method | LPIPS↓ | FID↓ | SSIM↑ | PSNR↑ | LPIPS↓ | FID↓ | SSIM↑ | PSNR↑ |
| **FFHQ 768×768** | | | | | | | | |
| ReSample | $0.457 \pm 0.087$ | 82.9 | $\mathbf{0.788 \pm 0.058}$ | $25.45 \pm 1.46$ | $0.366 \pm 0.053$ | 70.8 | $0.827 \pm 0.033$ | $21.83 \pm 1.68$ |
| FlowDPS | $0.431 \pm 0.117$ | 54.3 | $0.732 \pm 0.078$ | $27.64 \pm 2.20$ | $0.344 \pm 0.060$ | 42.5 | $0.771 \pm 0.048$ | $19.19 \pm 3.19$ |
| RSD | $0.458 \pm 0.098$ | 77.3 | $0.743 \pm 0.059$ | $27.67 \pm 2.47$ | $0.478 \pm 0.082$ | 73.3 | $0.736 \pm 0.048$ | $21.97 \pm 2.58$ |
| FlowChef | $0.406 \pm 0.093$ | 40.2 | $0.716 \pm 0.072$ | $25.81 \pm 2.61$ | $0.394 \pm 0.069$ | 69.8 | $0.780 \pm 0.051$ | $18.18 \pm 2.84$ |
| **Ours** | $\mathbf{0.236 \pm 0.070}$ | **10.7** | $0.772 \pm 0.055$ | $\mathbf{29.61 \pm 2.24}$ | $\mathbf{0.184 \pm 0.038}$ | **8.7** | $\mathbf{0.828 \pm 0.029}$ | $\mathbf{23.69 \pm 2.77}$ |
| **DIV2K 768×768** | | | | | | | | |
| ReSample | $0.556 \pm 0.146$ | 79.7 | $0.617 \pm 0.134$ | $21.79 \pm 2.52$ | $0.285 \pm 0.073$ | 51.9 | $0.796 \pm 0.067$ | $22.68 \pm 1.84$ |
| FlowDPS | $0.558 \pm 0.153$ | 65.5 | $0.536 \pm 0.148$ | $21.88 \pm 3.02$ | $0.328 \pm 0.103$ | 29.2 | $0.692 \pm 0.112$ | $21.71 \pm 2.67$ |
| RSD | $0.638 \pm 0.156$ | 97.6 | $0.551 \pm 0.136$ | $22.10 \pm 3.07$ | $0.464 \pm 0.112$ | 63.9 | $0.678 \pm 0.077$ | $23.23 \pm 2.21$ |
| FlowChef | $0.561 \pm 0.123$ | 49.6 | $0.486 \pm 0.148$ | $19.90 \pm 3.06$ | $0.489 \pm 0.148$ | 58.3 | $0.659 \pm 0.131$ | $20.87 \pm 2.65$ |
| **Ours** | $\mathbf{0.315 \pm 0.107}$ | **21.1** | $\mathbf{0.653 \pm 0.121}$ | $\mathbf{24.44 \pm 3.05}$ | $\mathbf{0.163 \pm 0.053}$ | **11.0** | $\mathbf{0.815 \pm 0.054}$ | $\mathbf{23.75 \pm 2.74}$ |

## A.9  FLAIR in Pixel Space

Table 5: **Quantitative results** with 50 NFE and $\sigma_\nu = 0.5\%$ – In-painting and Super-resolution ($\times 8$).

| | Inpainting | | | | SR $\times 8$ | | | |
|---|---|---|---|---|---|---|---|---|
| Method | LPIPS↓ | FID↓ | SSIM↑ | PSNR↑ | LPIPS↓ | FID↓ | SSIM↑ | PSNR↑ |
| DDNM | $0.158 \pm 0.042$ | 26.9 | $0.732 \pm 0.037$ | $18.31 \pm 2.94$ | $0.199 \pm 0.052$ | 31.9 | $0.635 \pm 0.079$ | $23.59 \pm 1.64$ |
| DPS | $0.195 \pm 0.064$ | 30.2 | $0.689 \pm 0.077$ | $20.49 \pm 2.81$ | $0.172 \pm 0.058$ | 27.8 | $0.658 \pm 0.088$ | $24.59 \pm 2.04$ |
| MM | $0.161 \pm 0.054$ | 28.8 | $0.728 \pm 0.062$ | $20.59 \pm 3.37$ | $0.172 \pm 0.051$ | 29.1 | $0.669 \pm 0.083$ | $24.65 \pm 1.97$ |
| $\Pi$GDM | $0.195 \pm 0.064$ | 30.2 | $0.689 \pm 0.077$ | $20.49 \pm 2.81$ | $0.157 \pm 0.052$ | 26.5 | $0.677 \pm 0.084$ | $24.98 \pm 2.07$ |
| **FLAIR** | $\mathbf{0.097 \pm 0.035}$ | **14.2** | $\mathbf{0.831 \pm 0.031}$ | $\mathbf{21.87 \pm 2.66}$ | $\mathbf{0.143 \pm 0.039}$ | 22.9 | $\mathbf{0.712 \pm 0.076}$ | $\mathbf{25.93 \pm 1.96}$ |

We additionally implement our method including DTA and $\lambda_{\mathcal{R}(t)} = \mathcal{L}_{CFM,t}^{-1}$ (0 for $t < 0.2$) in pixel space using the flow model from [33], trained on CelebA-HQ resized (256x256). For comparison, we rephrase score based baselines to flow following [26] and evaluate all on 1000 samples from the dataset on super-resolution and inpainting. The methods are hyperparameter-tuned to DDNM [53] (likelihood weight 4 for inpainting | 1 for SR8), DPS [11] (64 | 512), Moment Matching [43] (4 | 8), ΠGDM [50] (64 | 8) and pixel space FLAIR (0.5 | 32 and regularizer weight 0.4). As shown in Table 5, our method outperforms previous works also in pixel space, demonstrating its broader applicability.

## A.10  Runtime Analysis

We compare the runtime and memory consumption of our method to the baselines. As our hard data consistency can strongly influence the runtime, we also provide measurements with the number of data term steps $\leq 5$ and additionally a fast version using a "tinyVAE" of SD3. To validate that the

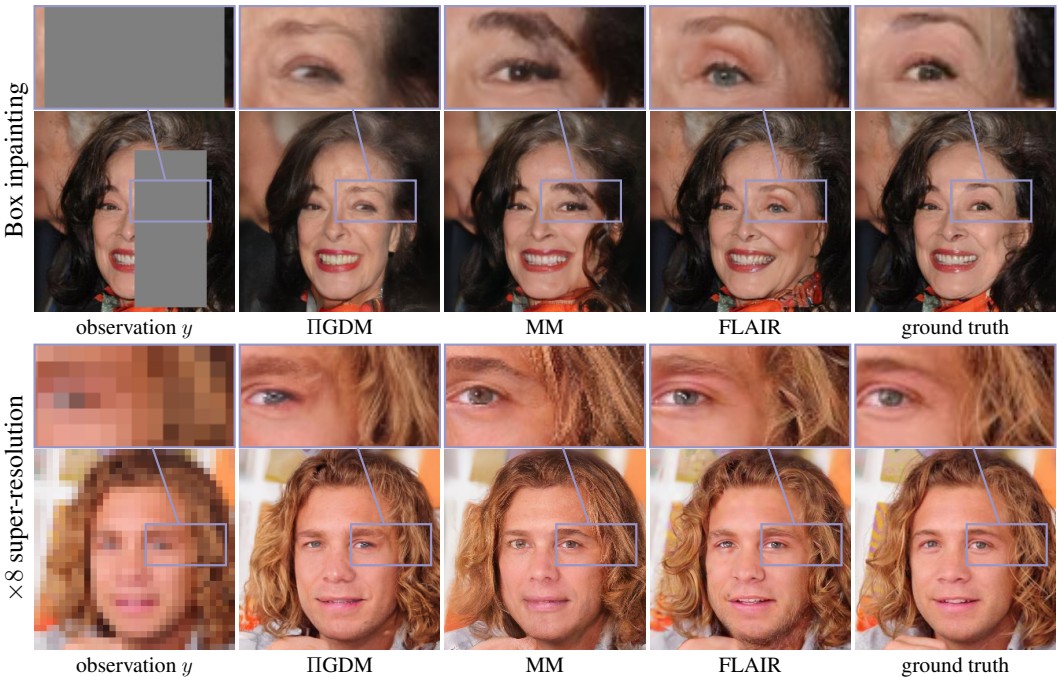

Figure 2: **Qualitative comparison.** FLAIR in pixel space produces posterior samples.

usage of the tinyVAE or less steps does not degrades the performance noticeably we also provide a metrics for x12 Super Resolution on 100 samples of FFHQ:

Table 6: Comparison of different methods in terms of runtime and memory usage. We validate the use of less data steps and a "tinyVAE" on $\times 12$ super resolution on 100 samples of the FFHQ dataset.

| Method | Runtime (s) $\downarrow$ | Memory (MB) $\downarrow$ | LPIPS $\downarrow$ | PSNR $\uparrow$ |
|---|---|---|---|---|
| Resample | 88.02 | 19009.2 | 0.461 | 25.31 |
| FlowDPS | 34.15 | 12228.6 | 0.404 | 27.74 |
| RSD (no repulsion) | 21.19 | 12400.0 | 0.462 | 28.11 |
| FlowChef | 15.23 | 12227.72 | 0.361 | 26.56 |
| **FLAIR (HDC, large VAE)** | 172.34 | 12389.4 | 0.259 | 27.42 |
| **FLAIR (HDC, tiny VAE)** | 40.77 | 5960.2 | 0.256 | 27.59 |
| **FLAIR (5 data term steps, tiny VAE)** | 22.46 | 5960.2 | 0.264 | 27.61 |

### A.11 Ensembling Experiment

To highlight that our model can also be used to obtain good MMSE estimates, we also conducted ensemble predictions by running posterior sampling 8 times and averaging the result. The results show that ensembling increases PSNR values, but reduces LPIPS and confirms that our samples are indeed distributed around the posterior mean and that results very close to the posterior mean like the baseline methods are perceptually further away (LPIPS) from the ground truth compared to our samples.

### A.12 Statistical Relevance

Our method is training-free, and the variance in reconstructed images is **intentional**, reflecting the stochasticity of our sampling process rather than instability. All methods are evaluated with identical random seeds to ensure fair comparison. We compute metrics over 1000 samples for FFHQ and 800 samples for DIV2K. Perceptual FID (pFID) is evaluated on $256 \times 256$ patches, resulting in 9000 and 7200 samples, respectively. Table 3 in Appendix A.8 reports means and standard deviations over multiple samples.

Table 7: **Quantitative results** – Super-resolution ($\times 8$ and $\times 12$). We report PSNR$\uparrow$ and LPIPS$\downarrow$. For ensembling we averaged 8 independent predictions of the corresponding methods. It can be seen that PSNR improves for all methods when ensembling. However, FLAIR shows the biggest gain, which means that our samples are indeed distributed around the mean and feature a higher variance compared to the baselines.

| | SR $\times 8$ | | SR $\times 12$ | |
| Method | PSNR$\uparrow$ | LPIPS$\downarrow$ | PSNR$\uparrow$ | LPIPS$\downarrow$ |
|---|---|---|---|---|
| **DIV2K** | | | | |
| FlowDPS | 22.53 | 0.4837 | 21.47 | 0.5524 |
| FlowDPS (8x ensemble) | 23.28 | 0.5157 | 22.06 | 0.5995 |
| FlowChef | 21.44 | 0.4898 | 20.20 | 0.5228 |
| FlowChef (8x ensemble) | 22.60 | 0.5502 | 21.60 | 0.5931 |
| **FLAIR** | 22.83 | **0.3627** | 21.05 | **0.4270** |
| **FLAIR (8x ensemble)** | **23.79** | 0.4244 | **22.27** | 0.4930 |
| **FFHQ** | | | | |
| FlowDPS | 29.02 | 0.3659 | 27.74 | 0.4036 |
| FlowDPS (8x ensemble) | 30.12 | 0.3267 | 28.65 | 0.3749 |
| FlowChef | 28.05 | 0.3303 | 26.56 | 0.3609 |
| FlowChef (8x ensemble) | 29.54 | 0.3267 | 28.15 | 0.3602 |
| **FLAIR** | 29.36 | **0.2028** | 27.42 | **0.2594** |
| **FLAIR (8x ensemble)** | **30.94** | 0.2457 | **29.00** | 0.2999 |

We also evaluated 100 FFHQ samples and 80 DIV2K samples, sampling three reconstructions per input for each method. We report the mean of each metric across all samples and the standard deviation of the means.

Table 8: Statistical evaluation on **FFHQ** for $\times$**8 Super Resolution**. We report mean $\pm$ standard deviation over 3 reconstructions per input.

| **Method** | LPIPS $\downarrow$ | FID $\downarrow$ | SSIM $\uparrow$ | PSNR $\uparrow$ |
|---|---|---|---|---|
| FlowDPS | 0.370 $\pm$ 0.0012 | 70.7 $\pm$ 1.45 | 0.755 $\pm$ 0.0010 | 28.98 $\pm$ 0.008 |
| RSD | 0.4678 $\pm$ 0.0001 | 102.9 $\pm$ 0.05 | 0.7362 $\pm$ 0.0001 | 28.45 $\pm$ 0.001 |
| FlowChef | 0.3316 $\pm$ 0.0055 | 63.5 $\pm$ 1.45 | 0.7593 $\pm$ 0.0027 | 28.12 $\pm$ 0.072 |
| Ours | **0.2039** $\pm$ 0.0048 | **40.5** $\pm$ 0.94 | **0.7970** $\pm$ 0.0228 | **29.74** $\pm$ 0.668 |

Table 9: Statistical evaluation on **FFHQ** for $\times$**12 Super Resolution**. We report mean $\pm$ standard deviation over 3 reconstructions per input.

| **Method** | LPIPS $\downarrow$ | FID $\downarrow$ | SSIM $\uparrow$ | PSNR $\uparrow$ |
|---|---|---|---|---|
| FlowDPS | 0.4073 $\pm$ 0.0002 | 77.6 $\pm$ 1.05 | 0.7391 $\pm$ 0.0006 | 27.71 $\pm$ 0.016 |
| RSD | 0.5039 $\pm$ 0.0002 | 119.0 $\pm$ 0.12 | 0.7217 $\pm$ 0.0001 | 27.08 $\pm$ 0.001 |
| FlowChef | 0.3626 $\pm$ 0.0050 | 81.1 $\pm$ 1.03 | 0.7283 $\pm$ 0.0027 | 26.62 $\pm$ 0.059 |
| Ours | **0.2593** $\pm$ 0.0023 | **45.8** $\pm$ 1.51 | **0.7582** $\pm$ 0.0252 | **27.81** $\pm$ 0.660 |

### A.12.1 t-Test Analysis

We further performed paired **t-tests** on the LPIPS scores between FlowDPS and FLAIR. The null hypothesis states that the mean LPIPS scores are the same for both methods. In all settings, we reject the null hypothesis ($p < 0.001$), confirming the statistical significance of our improvements see Table 16.

Table 10: Statistical evaluation on **FFHQ** for **Motion Blur**. We report mean ± standard deviation over 3 reconstructions per input.

| Method | LPIPS ↓ | FID ↓ | SSIM ↑ | PSNR ↑ |
|--------|---------|-------|--------|--------|
| FlowDPS | $0.4140 \pm 0.0030$ | $83.83 \pm 1.00$ | $0.7383 \pm 0.0006$ | $27.47 \pm 0.05$ |
| RSD | $0.4515 \pm 0.0001$ | $108.75 \pm 0.08$ | $0.7437 \pm 0.0001$ | $27.40 \pm 0.00$ |
| FlowChef | $0.4019 \pm 0.0007$ | $74.89 \pm 0.69$ | $0.7178 \pm 0.0022$ | $25.50 \pm 0.04$ |
| Ours | $\mathbf{0.2196} \pm 0.0080$ | $\mathbf{38.8} \pm 2.25$ | $\mathbf{0.7964} \pm 0.0319$ | $\mathbf{30.10} \pm 0.96$ |

Table 11: Statistical evaluation on **FFHQ** for **Inpainting**. We report mean ± standard deviation over 3 reconstructions per input.

| Method | LPIPS ↓ | FID ↓ | SSIM ↑ | PSNR ↑ |
|--------|---------|-------|--------|--------|
| FlowDPS | $0.3315 \pm 0.0015$ | $74.00 \pm 0.58$ | $0.7755 \pm 0.0007$ | $19.06 \pm 0.11$ |
| RSD | $0.4601 \pm 0.0003$ | $103.02 \pm 0.03$ | $0.7430 \pm 0.0000$ | $22.19 \pm 0.01$ |
| FlowChef | $0.3771 \pm 0.0013$ | $102.22 \pm 0.26$ | $0.7888 \pm 0.0016$ | $18.42 \pm 0.25$ |
| Ours | $\mathbf{0.1761} \pm 0.0012$ | $\mathbf{33.23} \pm 2.02$ | $\mathbf{0.8423} \pm 0.0172$ | $\mathbf{24.07} \pm 0.80$ |

## A.13 Additional Qualitative Examples

To illustrate the visual differences behind the error metrics, we present additional qualitative results for both FFHQ and DIV2k, comparing FLAIR with existing approaches. These examples complement the images in the main paper and highlight the visual fidelity, consistency, and robustness of our method across diverse scenes and different degradations. Figure 9 features a full sized version of the variance figure in section subsection 5.2.

## A.14 Failure cases

We observe two main failure modes for FLAIR, see Figure 10. First, we find that super-resolution on DIV2k occasionally results in grainy textures, usually in regions with abundant high-frequency detail and complicated light transport. Potentially, this happens for images which do not have high probability under the prior. We do not observe those artifacts for the FFHQ dataset. Second, we observe a few instances where the strong generative prior hallucinates semantically inconsistent or misaligned structures – especially facial features.

Table 12: Statistical evaluation on **DIV2K** for ×8 **Super Resolution**. We report mean ± standard deviation over 3 reconstructions per input.

| Method | LPIPS ↓ | FID ↓ | SSIM ↑ | PSNR ↑ |
|---|---|---|---|---|
| FlowDPS | $0.5517 \pm 0.0046$ | $138.24 \pm 1.67$ | $0.5207 \pm 0.0022$ | $22.41 \pm 0.02$ |
| RSD | $0.7163 \pm 0.0002$ | $181.83 \pm 0.15$ | $0.4892 \pm 0.0001$ | $21.99 \pm 0.00$ |
| FlowChef | $0.5726 \pm 0.0046$ | $145.77 \pm 3.31$ | $0.4998 \pm 0.0014$ | $21.21 \pm 0.04$ |
| Ours | $\mathbf{0.3716} \pm 0.0161$ | $\mathbf{88.08} \pm 1.43$ | $\mathbf{0.5991} \pm 0.0192$ | $\mathbf{23.06} \pm 0.41$ |

Table 13: Statistical evaluation on **DIV2K** for ×12 **Super Resolution**. We report mean ± standard deviation over 3 reconstructions per input.

| Method | LPIPS ↓ | FID ↓ | SSIM ↑ | PSNR ↑ |
|---|---|---|---|---|
| FlowDPS | $0.6264 \pm 0.0045$ | $154.31 \pm 0.43$ | $0.4866 \pm 0.0024$ | $21.37 \pm 0.02$ |
| RSD | $0.7714 \pm 0.0002$ | $198.85 \pm 0.12$ | $0.4683 \pm 0.0001$ | $21.15 \pm 0.00$ |
| FlowChef | $0.6020 \pm 0.0060$ | $151.32 \pm 2.45$ | $0.4586 \pm 0.0020$ | $20.10 \pm 0.03$ |
| Ours | $\mathbf{0.4316} \pm 0.0151$ | $\mathbf{101.12} \pm 4.51$ | $\mathbf{0.5236} \pm 0.0229$ | $\mathbf{21.35} \pm 0.51$ |

Table 14: Statistical evaluation on **DIV2K** for **Motion Deblur**. We report mean ± standard deviation over 3 reconstructions per input.

| Method | LPIPS ↓ | FID ↓ | SSIM ↑ | PSNR ↑ |
|---|---|---|---|---|
| FlowDPS | $0.6242 \pm 0.0082$ | $161.57 \pm 3.90$ | $0.4978 \pm 0.0028$ | $21.41 \pm 0.04$ |
| RSD | $0.8067 \pm 0.0002$ | $216.37 \pm 0.36$ | $0.4364 \pm 0.0001$ | $20.82 \pm 0.00$ |
| FlowChef | $0.6292 \pm 0.0007$ | $158.08 \pm 3.03$ | $0.4557 \pm 0.0025$ | $19.62 \pm 0.05$ |
| Ours | $\mathbf{0.3069} \pm 0.0036$ | $\mathbf{77.77} \pm 1.89$ | $\mathbf{0.6596} \pm 0.0316$ | $\mathbf{24.46} \pm 0.71$ |

Table 15: Statistical evaluation on **DIV2K** for **Inpainting**. We report mean ± standard deviation over 3 reconstructions per input.

| Method | LPIPS ↓ | FID ↓ | SSIM ↑ | PSNR ↑ |
|---|---|---|---|---|
| FlowDPS | $0.3738 \pm 0.0032$ | $106.58 \pm 0.89$ | $0.6579 \pm 0.0009$ | $21.06 \pm 0.05$ |
| RSD | $0.4667 \pm 0.0003$ | $136.82 \pm 0.21$ | $0.6650 \pm 0.0001$ | $23.07 \pm 0.00$ |
| FlowChef | $0.5111 \pm 0.0019$ | $128.97 \pm 0.60$ | $0.6355 \pm 0.0006$ | $20.51 \pm 0.02$ |
| Ours | $\mathbf{0.1729} \pm 0.0014$ | $\mathbf{51.41} \pm 0.48$ | $\mathbf{0.8122} \pm 0.0124$ | $\mathbf{24.06} \pm 0.87$ |

Table 16: Paired t-test $p$-values for LPIPS (FlowDPS vs. Ours). All comparisons are statistically significant.

| Dataset | Task | p-value |
|---|---|---|
| DIV2K | SR×8 | $2.92 \times 10^{-4}$ |
| | SR×12 | $7.19 \times 10^{-4}$ |
| | Motion Deblur | $5.30 \times 10^{-4}$ |
| | Inpainting | $1.21 \times 10^{-4}$ |
| FFHQ | SR×8 | $7.05 \times 10^{-5}$ |
| | SR×12 | $6.56 \times 10^{-5}$ |
| | Motion Deblur | $4.29 \times 10^{-5}$ |
| | Inpainting | $3.00 \times 10^{-5}$ |

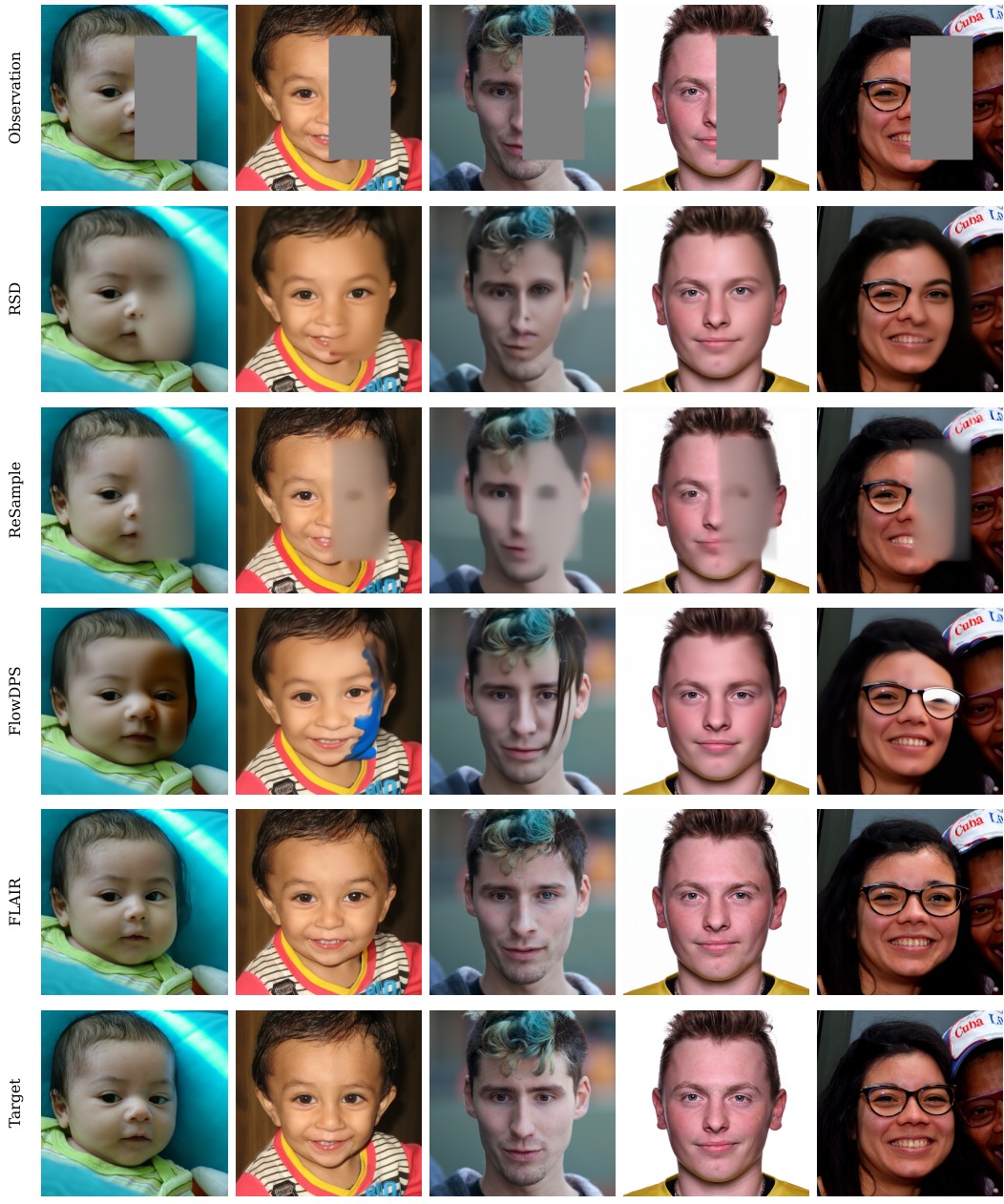

Figure 3: Inpainting results on FFHQ. Shown are observation, reference methods, FLAIR and ground truth. FLAIR produces realistic, high-frequency details while previous works either fail to inpaint the region correctly or collapse to overly smooth solutions.

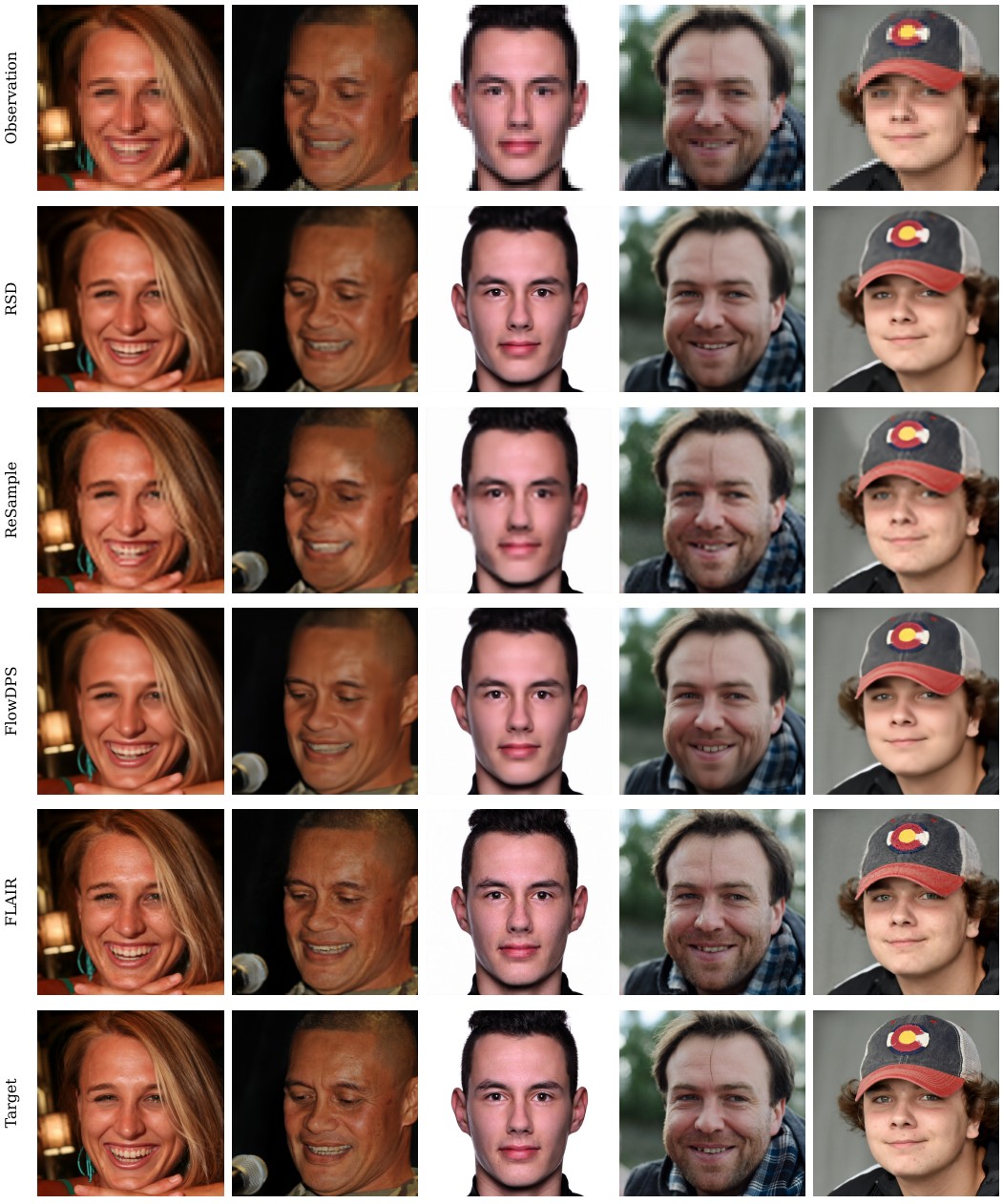

Figure 4: ×12 super-resolution results on FFHQ. Shown are observation, reference methods, FLAIR and ground truth. FLAIR produces sharp and results which still fulfill the data term, whereas the baselines tend to predict blurry images.

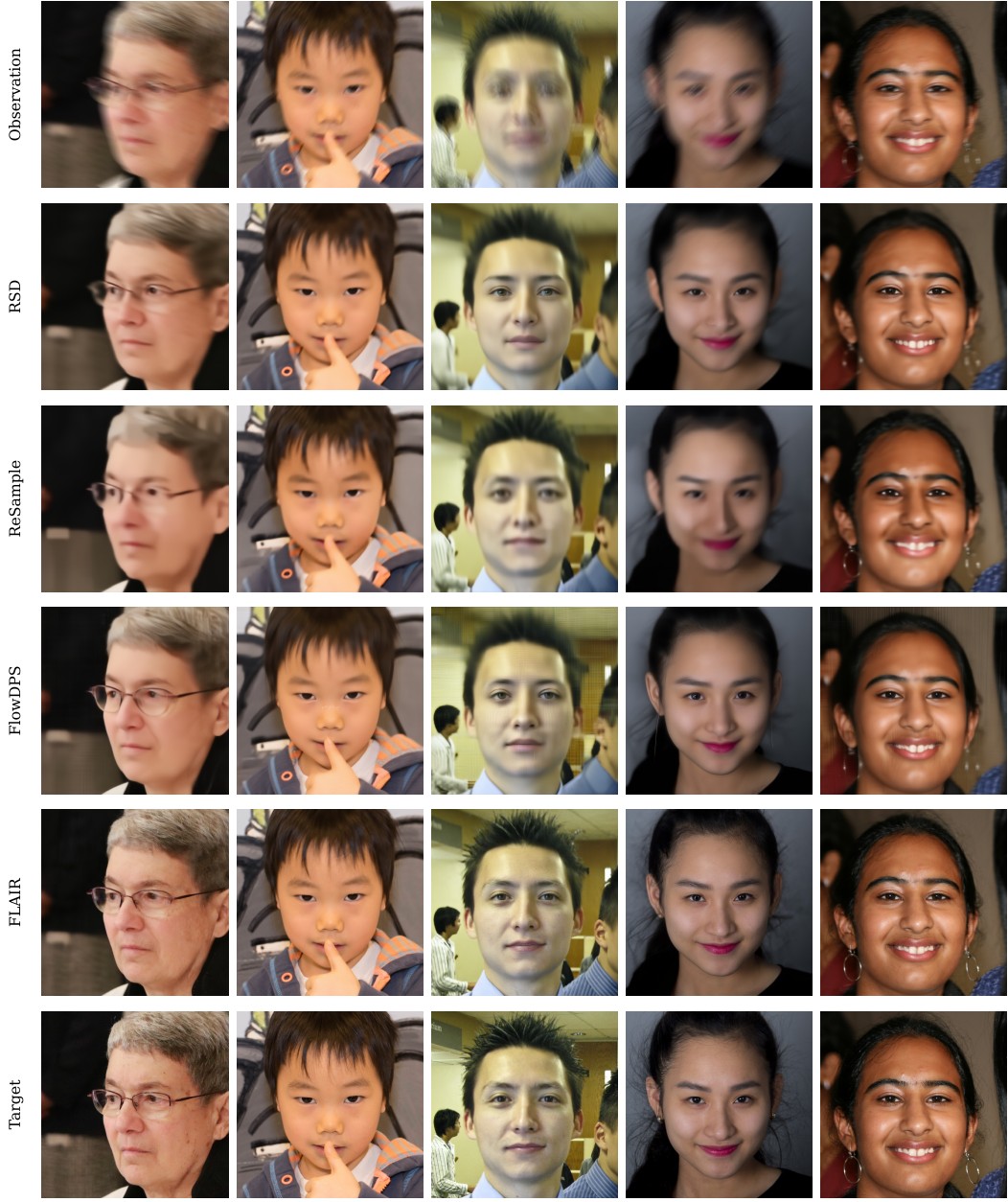

Figure 5: Motion de-blur results on FFHQ. Shown are observation, reference methods, FLAIR and ground truth. FLAIR produces sharp and results which still fulfill the data term, whereas the baselines tend to predict blurry images.

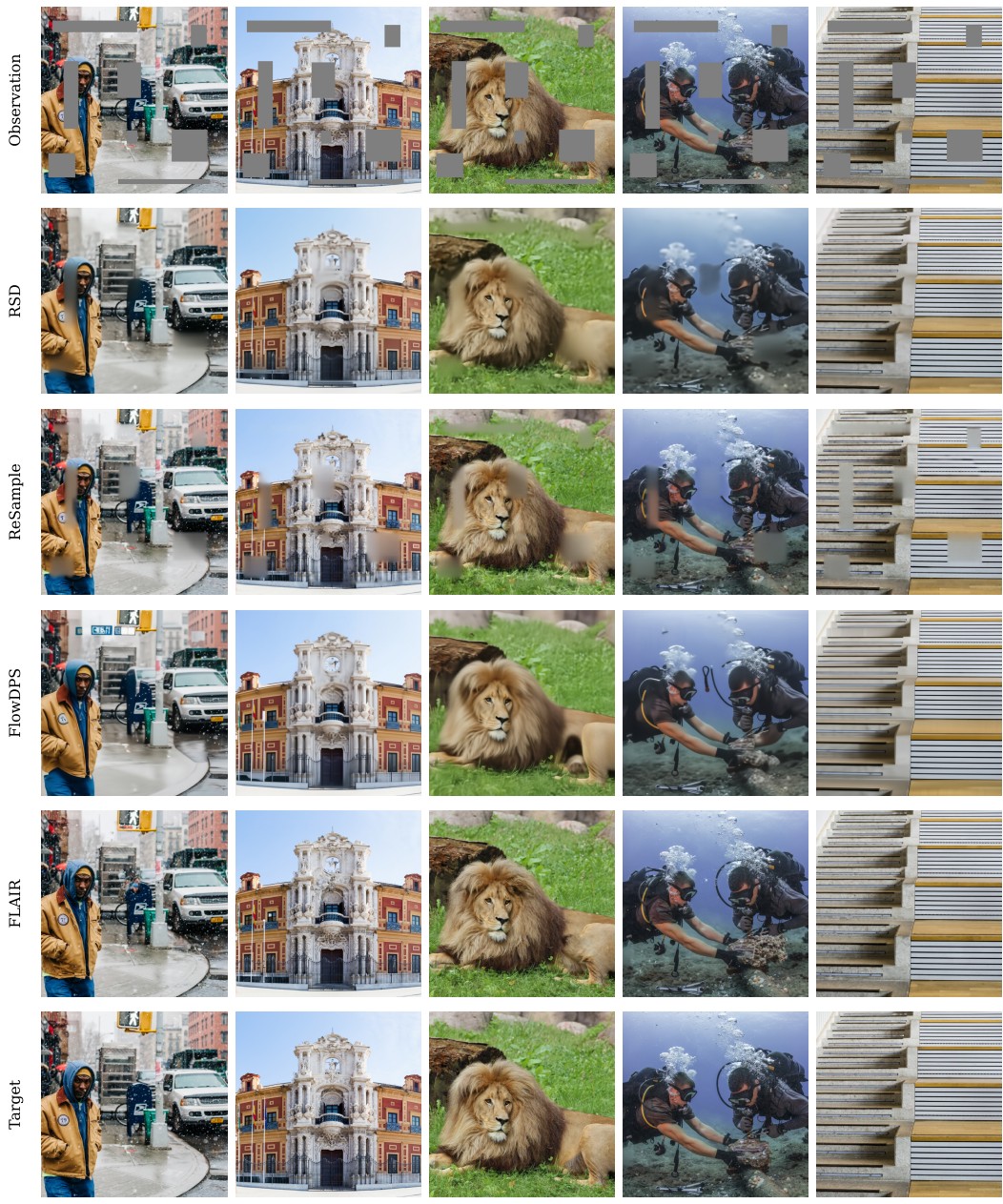

Figure 6: Inpainting results on DIV2k. Shown are observation, reference methods, FLAIR and ground truth. FLAIR produces realistic, high-frequency details while previous works either fail to inpaint the region correctly or collapse to overly smooth solutions. Moreover they do not fit the data term (not inpainted region) very well.

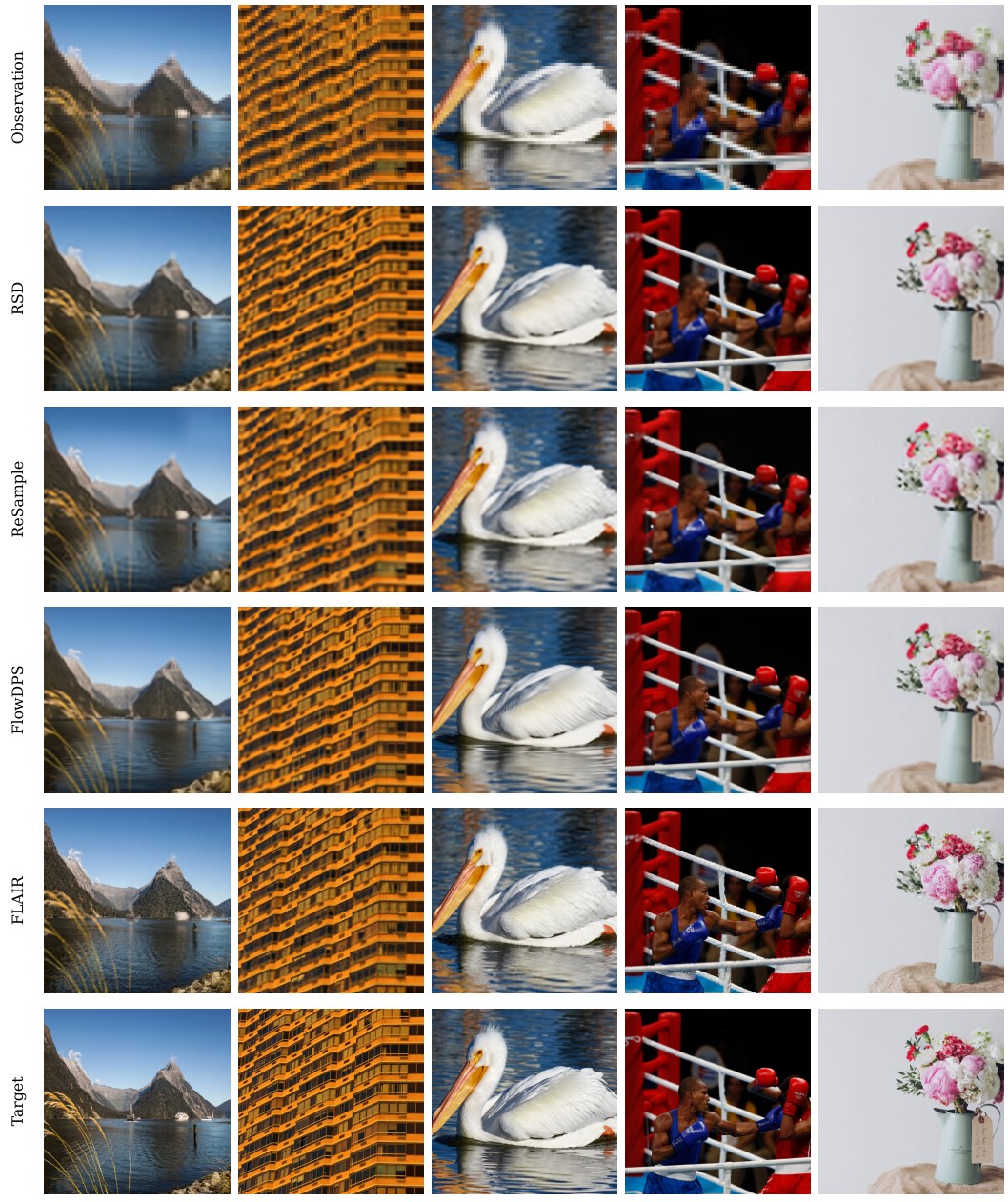

Figure 7: ×12 super-resolution results on DIV2k. Shown are observation, reference methods, FLAIR and ground truth. FLAIR produces sharp and results which still fulfill the data term, whereas the baselines tend to predict blurry images.

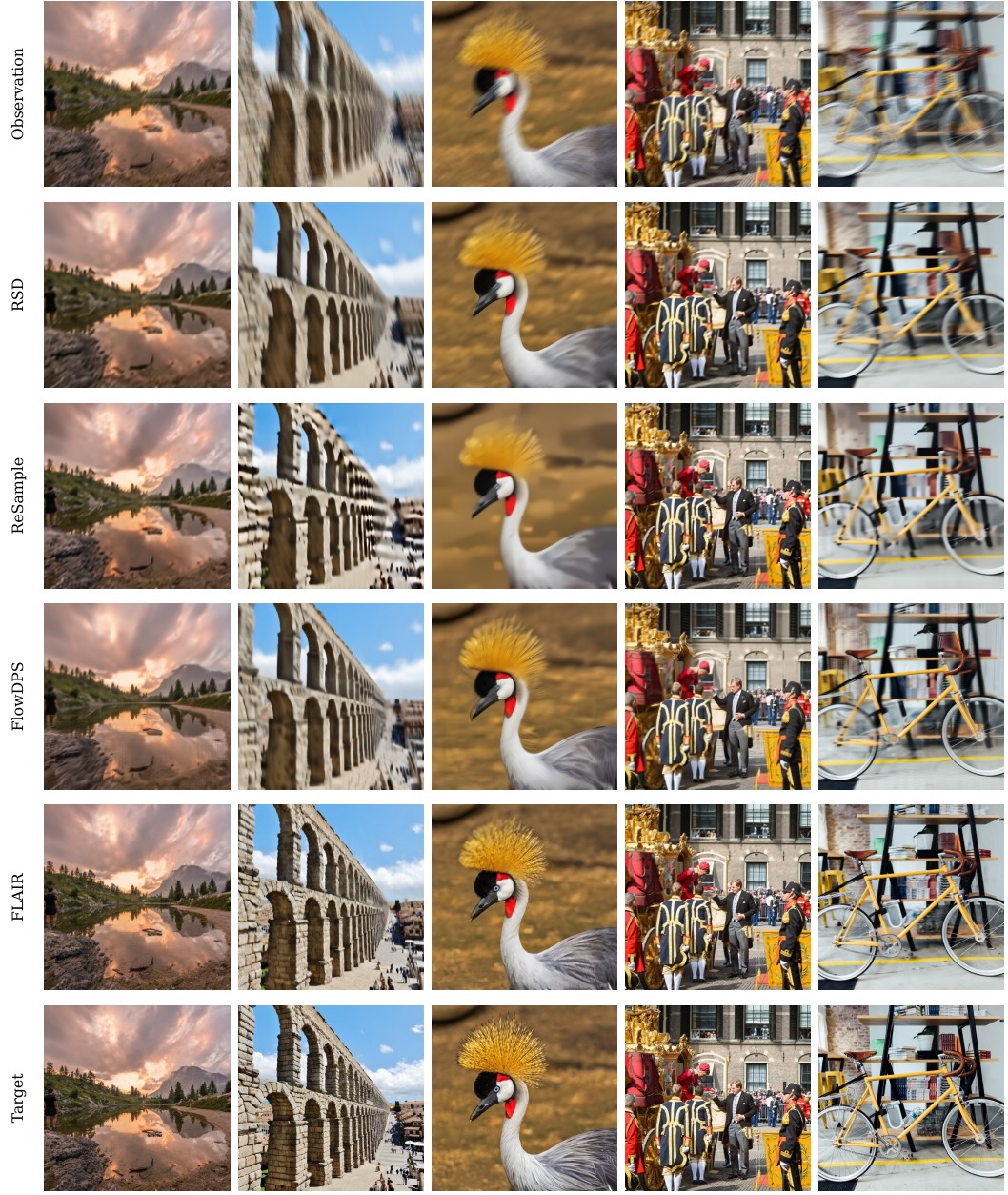

Figure 8: Motion de-blur results results on DIV2k. Shown are observation, reference methods, FLAIR and ground truth. FLAIR produces sharp and results which still fulfill the data term, whereas the baselines tend to predict blurry images.

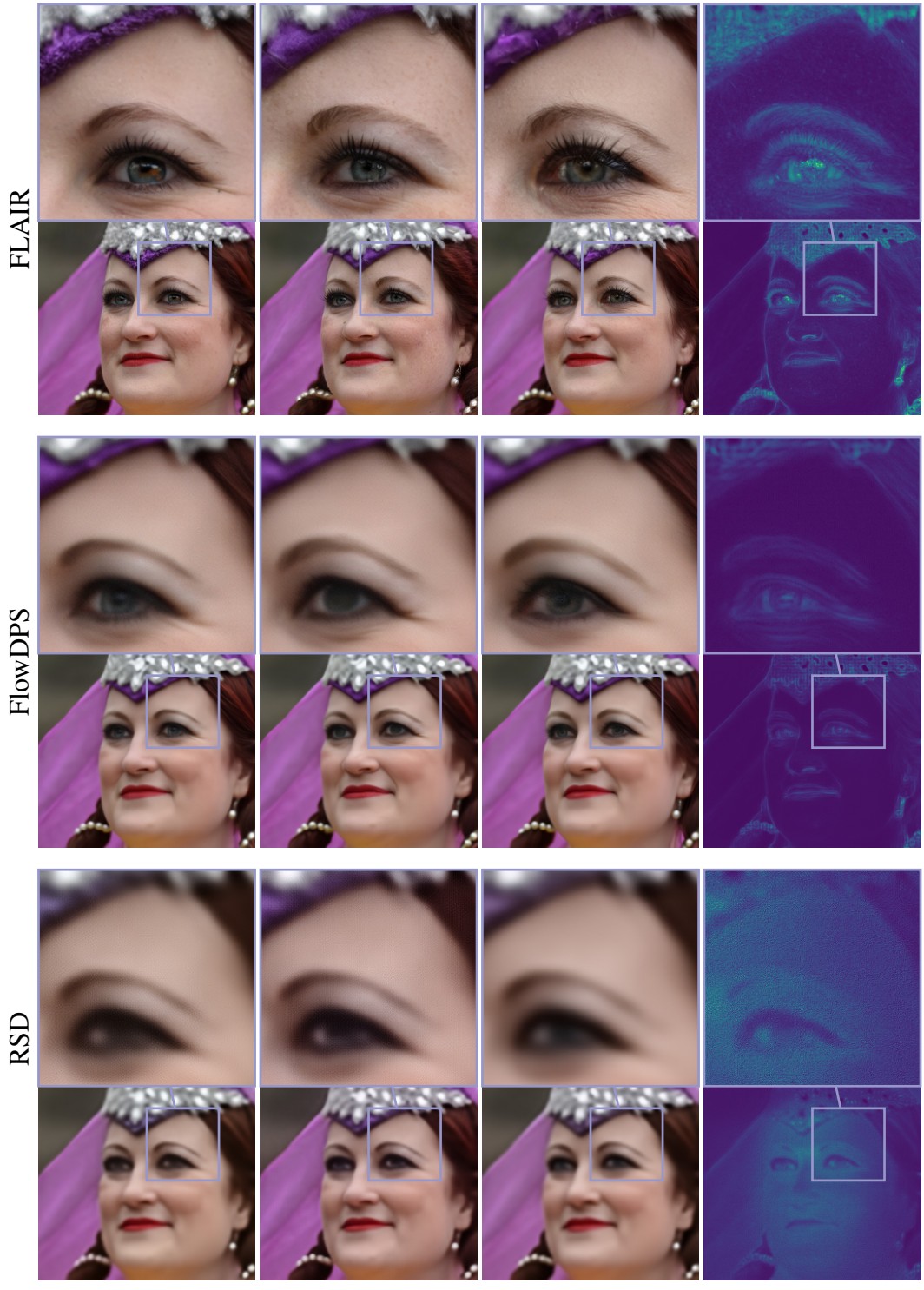

Figure 9: Individual samples for x12 Super Resolution with zoom and std. FLAIR produces varied samples from the posterior. For superresoltion The variance is expected to be mostly in the high frequencies, because the data term limits low frequency variations. The baselines tend to predict very similar looking images with less detail.

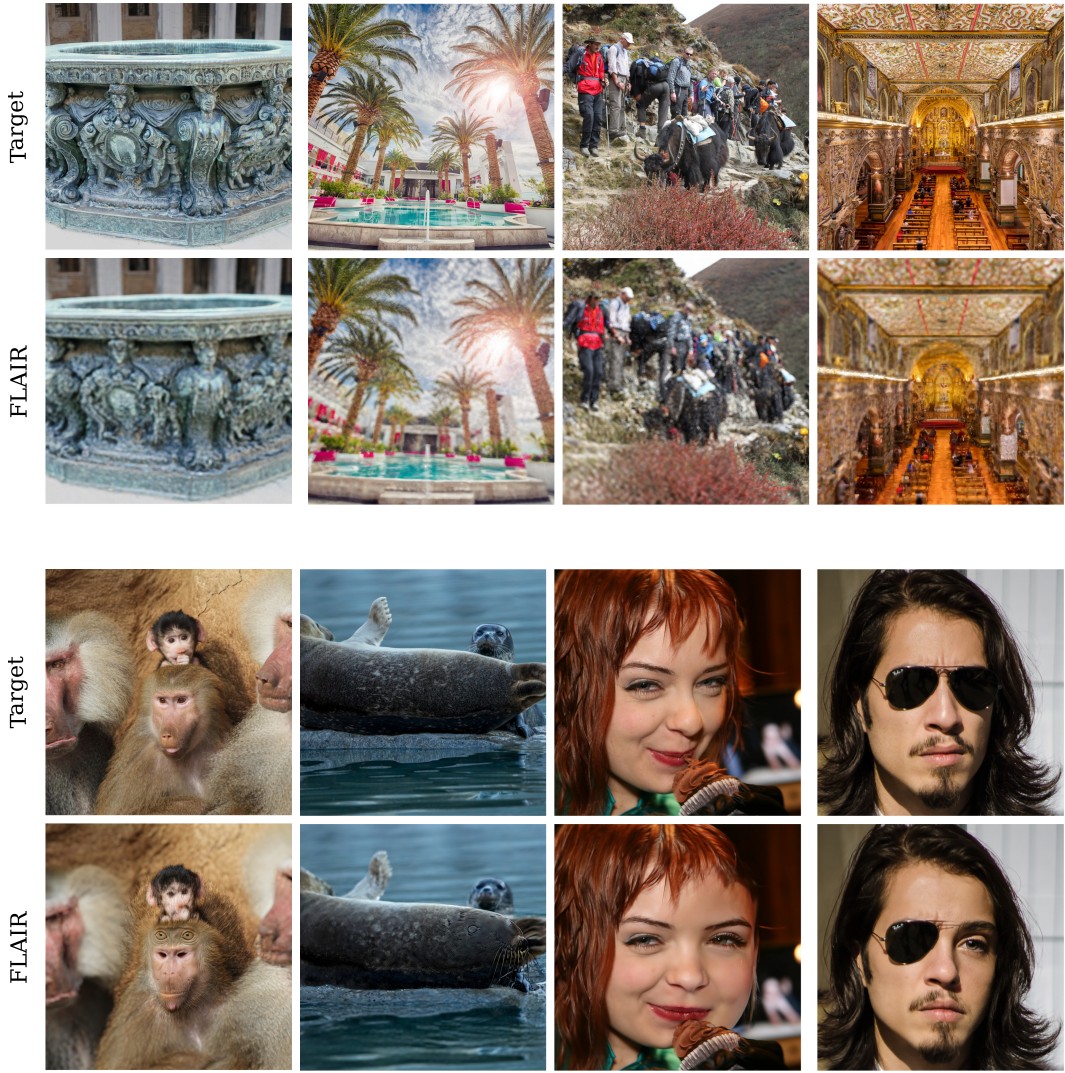

Figure 10: Qualitative failure cases of FLAIR on DIV2k and FFHQ. Top row: grainy results from systematic error. Those errors potentially stem from a weak prior for those images. For example we do not observe them for the FFHQ dataset Bottom row: Semantically inconsistent failures. Sometimes the model lacks the ability to incorporate globally consistent semantics into its restorations.

