# OpenReview forum: "Solving Inverse Problems with FLAIR"
_NeurIPS.cc/2025/Conference — NeurIPS 2025 poster_

### Official Review · Reviewer_fMnY · 2025-06-09

**Clarity:** 2
**Significance:** 2
**Originality:** 2
**Rating:** 4
**Confidence:** 3

**Summary:**

The authors introduce FLAIR, a training-free variational approach to exploit flow-based generative models as priors for solving inverse problems. In particular, the authors approximate the true posterior with a Gaussian distribution and optimise the mean of this Gaussian.

**Questions:**

I have a few questions and suggestions:
- As you deal with latent flow models, it would be good to explicitly have the Encoder/Decoder of the VAE as part of the Algorithm 1. Currently the notation is a bit inconsistent, in Algorithm 1 you write $\mu_x = A^T(y)$ and in line 146 you write $\mu_x = E(A^T y)$.
- In line 178 you write that you use stochastic gradient descent to enforce hard data consistency. What is stochastic for the optimisation problem $\arg \min_{\mu_x} \| y - A(\mu_x) \|^2$?
- You write in your abstract (line 13) "enforce exact consistency with the observed data". Given that you have noisy observation you almost never want to *exactly* fit the observations, because you do not want to overfit to the observation noise.
- At several points you claim that FLAIR is able to recover atypical modes (e.g., line 254). I do not fully see that the experiments support this claim. Which of the results corresponds to atypical modes?
- I think in Equation (8) the expectation over the time $t$ is missing (given the definition of $\mathcal{R}(x)$ in Equation (7) as a time integral.) .

Experiments:
- In Table 2 you obtain one of the highest PSNRs (both on FFHQ and DIV2K) if you do not use HDC, DTA and CRW. Why could this be the case?
- Ablation on Hard Data Consistency (Line 240). How do the observations $y$ influence the optimisation process (Algorithm 1) if you drop the HDC? Only through the initial input $\mu_x = A^T(y)$?
- What is $\nu = 0.01$ in Table 1? In Line 204 you write that you use an observation noise of $0.5$%
- I guess that the number of function evaluations of the forward operator $A$ and the Decoder of the VAE gets quite high, given that you solve $\arg \min_{\mu_x} \| y - A(\mu_x) \|^2$ in every iteration in Algorithm 1. Are there any ways to reduce the number of these function evaluations? If you would extend your method to other inverse problems (e.g., Computed Tomography) the evaluation of the forward operator $A$ can get quite expensive and this would be a bottleneck of the method.

**Ethical Concerns:**

["NO or VERY MINOR ethics concerns only"]

**Final Justification:**

Given the increasing use of large-scale flow-based models, extending inverse problem-solving techniques to this class is both relevant and important. The authors provided a thorough rebuttal, particularly w.r.t. computational efficiency (reducing the number of data term steps and using tiny VAE). Considering the rebuttal, as well as the responses to the other reviewers, I have raised my score to 4.

**Limitations:**

The authors address limitations.

**Quality:**

3

**Strengths And Weaknesses:**

### Strengths

The authors build on the variational approach to solving inverse problems with diffusion models, extending it to latent flow models. They propose a novel reparameterised variational distribution in Proposition 2. The paper is generally well-written (with exception of some minor inconsistencies) and clearly organized. Given the development of large scale latent flow models, i.e., StableDiffusion 3, the development of scalable methods to use these models as priors for inverse problems is important.

They show remarkable experiments on both FFHQ and DIV2K for several linear inverse problems and even show results for image editing. Also, the authors include detailed comparisons and ablations. Further, even showing failure cases (Figure 9 in the Appendix) is a good standard for transparency.


### Weaknesses

- My main issue is the similarity of FLAIR with RED-diff [1]. Several equations are exactly the same, e.g., Equation (7) in the manuscript and Equation (7) in [1] or Equation (8) in the manuscript and Lemma 1 in [1]. In my opinion the differences to RED-diff should be discussed more.
- In Line 111 you write that you set $\sigma_x^2 = 0$. However, for this choice the variational distribution $q(x_0 | y) = \mathcal{N}(\mu_x, \sigma_x^2)$ is not defined and the mathematical framework breaks down,
- I do not follow the argumentation for the posterior variance (starting line 223). Given that you set $\sigma_x^2 = 0$ (or i guess in the numerical experiments $\sigma_x^2 = $ some small constant?) you get an optimisation problem in Equation (5), where you approximate the true posterior with a Gaussian $\mathcal{N}(\mu_x, \sigma_x^2)$. However, it is not evident why optimizing Equation (5) multiple times would yield samples from the posterior. I would expect that you obtain the same mean $\mu_x$ everytime, given that you try to solve an optimisation task in Equation (5). Could you clarify the reasoning behind this claim?
- Due to the non-linear nature of the VAE every linear inverse problem in pixel-space becomes a non-linear inverse problem in latent space. Given that you are therefore always dealing with non-linear inverse problems, it would be interesting to also include a non-linear forward operator $A$ in pixel-space (e.g., non-uniform deblurring or phase retrieval).

Minor points:
- There are several inconsistencies regarding the notation. In Equation (2) you refer to the observations both with $z$ and with $y$. Same in Line 224. Further, you use both $A$ and $\mathcal{A}$ in Section 3.1.

[1] Mardani et al. "A variational perspective on solving inverse problems with diffusion models" (ICLR 2024)

---

> ### Author Rebuttal · Authors · 2025-07-28
>
> We thank the reviewer for acknowledging the novelty of our variational formulation, the relevance of our work in light of recent developments in large-scale latent diffusion models, the high quality of our results and the comprehensiveness of our experiments . Your feedback is much appreciated and we we incorporated it into our manuscript.
> If our answers are satisfying, we kindly ask to consider raising your score.
>
> **1. Similarity to RED-diff**
>
> The cited equations are part of our background section. We do not claim them as contribution, but they are important to understand the method. One of our contributions lies in the derivations of the variational objective for flow matching models stated in Propostion 1. The derivations can be found in the appendix A.1-A.3. We also compare to Repulsive Score Distillation (RSD), which adapts and extends RED-diff to latent diffusion models. We outperform RSD on all relevant metrics, which shows how valuable our contributions are.
> We will emphasize our contributions to the variational objective for flow matching models more clearly in the final version.
>
> **2. Zero variance of variational distribution**
>
> One can think of the new variational distribution as particle approximation of the posterior. In our experiments we only used one particle. With the particle approximation the gradients of the regularizer can be approximated using Wasserstein Gradient Flow and equation (7) is not required. However, eq. (7) motivates the weighting of the regularizer with the conditional flow matching loss for the CRW.
>
> **3. Source of Posterior Variance**
>
> While the variational objective remains fixed, noise vectors are drawn randomly during the optimization which leads to a different optimization trajectory. This is analogous to how different random seeds with the same text prompt yield diverse outputs in generative models like Stable Diffusion. In our case, these different trajectories correspond to different posterior modes.
>
> **4. Non linear forward operators**
>
> As you said the decoder, which is part of the forward operator for latent models makes all the tested tasks non-linear forward operators. We added the task of non-linear deblurring. The non-linearity lies in the non-linear conversion from sRGB to linear color space and back. The blur opoerator is applied in the linear color space. Additionally, values are clipped after blurring simulating saturation effects.
>
> | Method (FFHQ) | LPIPS ↓ | FID ↓ | SSIM ↑ | PSNR ↑ |
> | --- | --- | --- | --- | --- |
> | Resample | 0.446 | 118.1 | **0.792** | 25.21 |
> | FlowDPS | 0.413 | 83.0 | 0.738 | 27.27 |
> | RSD (no repulsion) | 0.448 | 109.87 | 0.7441 | 27.27 |
> | FlowChef | 0.402 | 74.5 | 0.716 | 25.29 |
> | Ours | **0.201** | **38.0** | 0.7851 | **29.37** |
>
> | Method (DIV2K) | LPIPS ↓ | FID ↓ | SSIM ↑ | PSNR ↑ |
> | --- | --- | --- | --- | --- |
> | Resample | 0.551 | 149.1 | 0.607 | 21.79 |
> | FlowDPS | 0.612 | 157.8 | 0.500 | 21.30 |
> | RSD (no repulsion) | 0.626 | 177.3 | 0.542 | 21.81 |
> | FlowChef | 0.632 | 160.4 | 0.455 | 19.57 |
> | Ours | **0.303** | **78.2** | **0.644** | **23.99** |
>
> **5. Stochasticity of HDC**
> Thank you for spotting this error. The optimization of the data term is not stochastic. The optimization of the variational objective however is due to the random sampling of noise vectors.
>
> **6. Overfitting to noise in HDC**
>
> The optimization of the data term can be stopped early for noisy observations as it is the case for our experiments. As mentioned in A.4 we stop at an error of less than $1\times 10^{-4} \cdot \text{len}(y)$. However, for tasks like inpainting, absence of measurement noise is realistic.
>
> **7. Claim that FLAIR is able to recover atypical modes**
>
> This is a conclusion we draw from the observation that FLAIR reconstructs diverse samples (see Fig. 3) which also look realistic. As shown in Fig. 3 baseline approaches collapse to a single solution which is close to the mean, but does not look realistic. Under the unconditional prior all modes are unlikely. One has to condition the model very strictly to increase the likelihood for certain modes e.g. with very precise text prompts which is not feasible for degraded images. Our solution is to compute “special noise vectors” $\hat{x}$ to condition the model and increase the likelihood of modes which match the measurement.
>
> **8. Expectation over time t**
>
> You are correct. Thank you very much for pointing out that error. We will corrected it in the manuscript.
>
> **9. Why does the model perform better without HDC, DTA, CRW?**
>
> At this point we also want to emphasize that PSNR and SSIM are not ideal metrics to access the quality of the reconstructions, because they favor a MMSE prediction which is overly smooth and unrealistic. Consequently, any plausible texture shift, while perceptually correct, incurs a large $\ell_2$ penalty and reduces PSNR/SSIM. LPIPS on the other hand is able to measure distances between images in a way that correlate with the human perception
>
> **When deactivating our contributions the method severely degrades terms of LPIPS**. The explanations is that the results collapse to a posterior mean similar to the baseline methods. This mean-prediction achieves a higher PSNR, but is perceptually farther from the ground truth and achieves worse LPIPS, as the respective qualitative results in the ablation study show.
>
> To highlight that our model can also be used to obtain good MMSE estimates, we also conducted ensemble predictions by running posterior sampling 8 times and averaging the result. The results show that ensembling increases PSNR values, but reduces LPIPS and confirms that our samples are indeed distributed around the posterior mean and that results very close to the posterior mean like the baseline methods are perceptually further away (LPIPS) from the ground truth compared to our samples. We included the ensembling experiments in our supplementary material.
>
> 80 samples of DIV2K
>
> | Method (DIV2K) | x8 PSNR | x8 LPIPS | x12 PSNR | x12 LPIPS |
> | --- | --- | --- | --- | --- |
> | Resample | 22.08 | 0.5480 | 20.60 | 0.6576 |
> | FlowDPS | 22.53 | 0.4837 | 21.47 | 0.5524 |
> | FlowDPS (8x ensemble) | 23.28 | 0.5157 | 22.06 | 0.5995 |
> | RSD | 23.45 | 0.5389 | 21.96 | 0.6836 |
> | FlowChef | 21.44 | 0.4898 | 20.20 | 0.5228 |
> | FlowChef (8x ensemble) | 22.60 | 0.5502 | 21.60 | 0.5931 |
> | Ours | 22.83 | 0.3627 | 21.05 | 0.4270 |
> | Ours (8x ensemble) | 23.79 | 0.4244 | 22.27 | 0.4930 |
>
> 100 samples of FFHQ
>
> | Method (FFHQ) | x8 PSNR | x8 LPIPS | x12 PSNR | x12 LPIPS |
> | --- | --- | --- | --- | --- |
> | Resample | 26.26 | 0.3860 | 25.31 | 0.4606 |
> | FlowDPS | 29.02 | 0.3659 | 27.74 | 0.4036 |
> | FlowDPS (8x ensemble) | 30.12 | 0.3267 | 28.65 | 0.3749 |
> | RSD | 29.69 | 0.3915 | 28.11 | 0.4624 |
> | FlowChef | 28.05 | 0.3303 | 26.56 | 0.3609 |
> | FlowChef (8x ensemble) | 29.54 | 0.3267 | 28.15 | 0.3602 |
> | Ours | 29.36 | 0.2028 | 27.42 | 0.2594 |
> | Ours (8x ensemble) | 30.94 | 0.2457 | 29.00 | 0.2999 |
>
> **10. HDC ablations settings**
>
> Without HDC we perform only 1 step on the data term similar to some of the baselines. We will clarify this in the manuscript.
>
> **11. What is $\nu=0.01$ in Table 1? In Line 204 you write that you use an observation noise of 0.5\%**
>
> Many methods specify the noise in absolute terms without mentioning the data range. We use 0.5\% of the data range as noise which corresponds to 0.01 on a range from -1 to 1. We will be consistent in the final version.
>
> **12. Reduction of NFEs**
>
> Our method can also be deployed with a reduced number of data term steps (e.g., maximum of 5), if a faster runtime is required. Furthermore, one can switch to a "tinyVAE" version of SD3.
>
> To validate that the usage of the tinyVAE or less steps does not degrades the performance noticeably we also provide a metrics for x12 Super Resolution on 100 samples of FFHQ:
>
> In our manuscript we set the number of NFEs to a fixed value for all methods and baselines for an easier comparison. For different tasks the required number might be lower or higher. While runtime optimization is out of scope for this project we want to highlight that the number of NFEs needed does not only depend on the posterior sampling method used but more so on the stochastic interpolant. For example RED-diff used 1000 NFEs for most of their experiments on a different model.
>
> | Method | Runtime (s) ↓ | Memory (MB) ↓ | LPIPS ↓ | PSNR↑ |
> | --- | --- | --- | --- | --- |
> | Resample | 88.02 | 19009.24 | 0.4606 | 25.31 |
> | FlowDPS | 34.15 | 12228.63 | 0.4036 | 27.74 |
> | RSD (no repulsion) | 21.19 | 12399.96 | 0.4624 | 28.11 |
> | FlowChef | 15.23 | 12227.72 | 0.3609 | 26.56 |
> | Ours (HDC, largeVAE) | 172.34 | 12389.37 | 0.259 | 27.42 |
> | Ours (HDC, tiny VAE) | 40.77 | 5960.22 | 0.256 | 27.59 |
> | Ours (5 data term steps, tiny VAE) | 22.46 | 5960.22 | 0.264 | 27.61 |

---

> > ### Comment · Reviewer_fMnY · 2025-08-01
> > **Response**
> >
> > Thank you for the rebuttal and the additional experiments.
> >
> > **3. Source of Posterior Variance**
> >
> > Yes, I understand that there is the additional noise input in every iteration (last line in Algorithm 1). However, I am not sure why this should lead to different modes of the posterior. I see in Figure 3 that you get a variety in solutions $\mu_x$, but I am not sure if these really correspond to samples from the posterior.
> >
> > **9. Why does the model perform better without HDC, DTA, CRW?**
> >
> > I assume that 12x PSNR is obtained by averaging over 12 runs of your method? Why are the results of 12x PSNR worse than for 8x PSNR?
> >
> > **Deterministic trajectory adjustments (DTA)**
> >
> > In the responses to the other reviewers you write about the DTA you propose. Can you maybe compare your DTA to the recently proposed methods in [1] and [2]? In my understanding they also deal include an interpolation step to project the current reconstruction back onto a "region of the learned prior that has reasonably high support/density" (line 158).
> >
> >
> > [1] Martin et al. "Pnp-flow: Plug-and-play image restoration with flow matching" (ICLR 2025)
> >
> > [2] Cheng et al. "Gradient-free generation for hard-constrained system" (ICLR 2025)

---

> > > ### Author Response · Authors · 2025-08-02
> > >
> > > Thank you for opening the discussion.
> > >
> > > **3. Source of Posterior Variance**
> > >
> > > For a multi-particle approximation for the variational distribution, the gradients for each particle only differ due to the random noise vector. Therefore different runs with a single particle also yield different optima, if different noise vectors are drawn during the optimisation. For imaging applications with highly multi modal posteriors we can find different modes by doing that. If there would only be 1 valid mode the optimisation should always give the same results. One could also use more sophisticated particle approximations. E.g. the RSD method we compared to use a repulsive term to add more variance. However, they observed that the repulsion reduces quality. Furthermore, we observed that the introduced variance often results in unrealistic particles (see Fig 3 4th RSD result).
> > >
> > > **9. Why does the model perform better without HDC, DTA, CRW?**
> > >
> > > We are sorry for the confusion. The tasks we evaluated on are x8 and x12 Super Resolution. That is why the x12 columns has worse results. We put the number of predictions behind the method name in the rows. For both tasks we used 8 predictions in the ensemble.
> > >
> > > **DTA compared to prior work**
> > >
> > > The cited methods simply add noise to the image according to the timestep t, because the denoising model requires inputs with a matching SNR and t. For optimal transport flow matching this renoising is an interpolation between the image and a noise vector.
> > >
> > > Our proposed DTA does not simply use a random noise vectors but a mix of random noise $/epsilon$ and a deterministic noise vector $/hat{x_1}$. The latter additionally conditions the model on the current estimate $/mu$ to guide the model in the right direction. Intuitively, without DTA the velocity prediction could point into directions that can not be explained by the measurement and hinder convergence which results in blurry results in practise.

---

> > > > ### Comment · Reviewer_fMnY · 2025-08-05
> > > > **Response**
> > > >
> > > > Thank you for the response. I think you have addressed all of my previous comments.
> > > >
> > > > One small clarification for **12. Reduction of NFEs**
> > > >
> > > > In your algorithm you add the hard data consistency $ \mu_x = \arg \min_x || A(x) - y ||_2^2$ and you are always using 5 steps (gradient descent?) steps to solve this. If you look at RED-diff (Algorithm 1) [1] they are only using one step (i.e. just the gradient). Does your work also work for in this setting?
> > > >
> > > > My point is that in some application the evaluation of the score model might actually be cheaper than the application of the forward operator $A$ (and its transpose $A^T$).
> > > >
> > > >
> > > > [1] Mardani et al. "A Variational Perspective on Solving Inverse Problems with Diffusion Models"

---

> > > > > ### Author Response · Authors · 2025-08-06
> > > > >
> > > > > Thank you for your question regarding the reduction of NFEs.
> > > > > Our method also performs well when the data consistency constraint is relaxed to a single gradient step on the data term. While this no longer enforces true hard data consistency, it offers a more efficient alternative with minimal performance loss. We include the corresponding results in the table below.
> > > > >
> > > > > | Method | Runtime (s) ↓ | Memory (MB) ↓ | LPIPS ↓ | PSNR ↑ |
> > > > > | --- | --- | --- | --- | --- |
> > > > > | Resample | 88.02 | 19009.24 | 0.4606 | 25.31 |
> > > > > | FlowDPS | 34.15 | 12228.63 | 0.4036 | 27.74 |
> > > > > | RSD (no repulsion) | 21.19 | 12399.96 | 0.4624 | 28.11 |
> > > > > | FlowChef | 15.23 | 12227.72 | 0.3609 | 26.56 |
> > > > > | Ours (HDC, largeVAE) | 172.34 | 12389.37 | 0.259 | 27.42 |
> > > > > | Ours (HDC, tiny VAE) | 40.77 | 5960.22 | 0.256 | 27.59 |
> > > > > | Ours (5 data term steps, tiny VAE) | 22.46 | 5960.22 | 0.264 | 27.61 |
> > > > > | Ours (1 data term step, tiny VAE) | 9.44 | 5960.22 | 0.3056 | 27.07 |

---

### Official Review · Reviewer_5yNE · 2025-07-02

**Clarity:** 4
**Significance:** 3
**Originality:** 2
**Rating:** 4
**Confidence:** 3

**Summary:**

The author proposes FLAIR, a scheme that combines latent generative modeling, flow matching, and variational inference into a unified solution for inverse problems. Their designed DTA (Deterministic Trajectory Adjustment) and CRW (Calibrated Regularizer Weight) modules improves the performance of flow based image generation priors in solving image inverse problems. The proposed framework and modules are interesting. The author tested the proposed method on image restoration, image super-resolution (8x and 12x), and image motion deblurring tasks, and demonstrated that FLAIR visually outperforms other existing methods.

**Questions:**

1. Overall, this study appears to present a progressive innovation building on existing work. To enhance clarity, the authors could consider reorganizing or more prominently emphasizing the core insight or innovation of their approach.

2. In abstract, the author proposes several challenges to existing methods, such as "the encoding into a lower dimensional latent space makes the underlying (forward) mapping non-linear" and "learned generative models struggle to recover rare, atypical data modes during inference". How does the FLAIR studies and addresses these challenges? What insights will it bring to this field?

3. Regarding methodology, clarify how the weighting parameters within the Calibrated Regularizer Weight module are determined. Are these parameter settings, along with task-specific variables (inpainting box size, zoom-in factors, kernel size, SNR), treated as independent?  Suggest the author to provide more explanations or proofs for optimizing convergence using Calibrated Regularizer Weight and Deterministic Trajectory Adjustment.

4. While FLAIR demonstrates strengths, its performance on PSNR/SSIM metrics does not currently show an advantage compared to other image generation methods; Suggest the author to clarify/explain this point. In addition, there is a lack of comparison with non-flow models (e.g., esrgan, deblurgan).

5. While noise is modeled as additive Gaussian (Line 78), real-world image inverse problems often involve non-Gaussian noise.

6. Can the author provide the training time of the prior models and the inference time on each task.

7. Given that prior work (e.g., Singanallur V. Venkatakrishnan et al., Plug-and-Play Priors for Model Based Reconstruction) has explored decoupling data fidelity and regularization via optimization algorithms, could the authors elaborate on how their hard data consistency module specifically differs from this established concept of decoupled optimization?

**Ethical Concerns:**

["NO or VERY MINOR ethics concerns only"]

**Final Justification:**

I have read the authors' rebuttals. The authors have addressed my concerns, including clarifying the innovation of this research relative to existing work and adding discussions on quantitative evaluation results and on non-Gaussian cases. Taking into account the innovation and contribution of this study, I have updated rating in the system.

**Limitations:**

In the Discussion section, the authors should also provide future optimization directions regarding the limitations.

**Paper Formatting Concerns:**

I do not raise formatting concerns.

**Quality:**

3

**Strengths And Weaknesses:**

The author proposed FLAIR, which provides a new solution for image inverse problem by combining flow-base image generative model priors, and achieve enhanced performance in visual. I am concerned about the innovativeness of this study and significance of experimental results (especially PSNR/SSIM metrics). See Questions and Limitations for detalis.

---

> ### Author Rebuttal · Authors · 2025-07-27
>
> We thank the reviewer for acknowledging the strong perceptual performance of our method. Your feedback is much appreciated and we incorporated it into our manuscript.
> If our answers are satisfying, we kindly ask to consider raising your score.
>
> Our work is positioned within the field of training-free posterior sampling using generative priors. Given that flow matching models represent the current state-of-the-art in image generation, we believe it is both timely and important to develop posterior sampling methods that are specifically compatible with these powerful generative priors.
>
> **1. Highlight insights and innovations**
>
> We have revised the manuscript to put more emphasis on clearly highlighting the core innovation of our approach.
>
> - **Variational posterior sampling for flow matching:**
>
>     To the best of our knowledge, we are the first to introduce a variational approach for posterior sampling with flow matching models. The derivation of this method is non-trivial and is provided in detail in Appendix A.1–A.3. We also compare to Repulsive Score Distillation (RSD), which adapts and extends RED-diff to latent diffusion models. We outperform RSD on all relevant metrics, which shows how valuable our contributions are.
>
> - **Deterministic trajectory adjustments:**
>
>     Our proposed deterministic trajectory adjustment is a crucial and novel component of the method, and our ablation studies show it yields the largest improvements in LPIPS. This adjustment is theoretically motivated and, to our knowledge, has not been previously explored in the literature.
>
> - **Model accuracy-weighted regularization (CRW):**
>
>     We are the first to propose using model accuracy as a weight for the regularization term (CRW), which is feasible only for variational posterior sampling approaches. This is not possible for non-variational methods.
>
> - **Hard data consistency in a variational framework:**
>
>     While hard data consistency has been used in other works, our framework (FLAIR) is, to our knowledge, the first to apply it within a variational setting.
>
>
> **2. Challenges from the abstract**
>
> Our method addresses non-linear mappings by operating in the latent space while explicitly backpropagating through the decoder to ensure consistency with the observed data in pixel space. This allows us to handle the non-linear encoder–decoder structure effectively, while still being able to initialze with the pseudo inverse.
> We observed that FLAIR reconstructs diverse samples (see Fig. 3) which also look realistic. As shown in Fig. 3 baseline approaches collapse to a single solution which is close to the mean, but does not look realistic. Under the unconditional prior all modes are unlikely. One has to condition the model very strictly to increase the likelihood for certain modes e.g. with very precise text prompts which is not feasible for degraded images. Our solution is to compute “special noise vectors” $\hat{x}$ to condition the model and increase the likelihood of modes which match the measurement.
>
> **3. CRW weighting**
>
> The CRW weighting is determined through a one-time accuracycalibration procedure, independent of the task or any hyperparameter. Specifically, we compute the expected model error using a small set of arbitrary calibration images. This error is derived from the conditional flow-matching loss, which directly corresponds to the regularization term in our framework. Additional details can be found in the Appendix A.6.
>
> **4. PSNR/SSIM metrics**
>
> We want to **highly emphasize** that our goal is to obtain diverse and visually convincing reconstructions that remain consistent with the observed data (i.e., sampling from the posterior modes), rather than pixel-wise accurate ones (i.e., estimating the posterior mean). This inherently leads to lower PSNR and SSIM scores, since there are many possible samples that can explain the observed data. However, our method achieves significantly closer reconstruction in perceptual distance as shown by the LPIPS evaluations. Furthermore, the strong FID scores show that our results are highly realistic while baselines fail to reconstruct images which look realistic.
>
> Due to the perception-distortion tradeoff, it is fundamentally not possible to achieve both high perceptual quality and high pixel-level fidelity simultaneously, especially under severe degradations. Reconstructions that appear realistic often deviate from the pixel-wise mean and therefore yield lower PSNR and SSIM, despite being perceptually closer to a human observer.
>
> To highlight that our model can also be used to obtain good MMSE estimates, we also conducted ensemble predictions by running posterior sampling 8 times and averaging the result. The results show that ensembling increases PSNR values, but reduces LPIPS and confirms that our samples are indeed distributed around the posterior mean and that results very close to the posterior mean like the baseline methods are perceptually further away (LPIPS) from the ground truth compared to our samples. We included the ensembling experiments in our supplementary material.
>
> 100 samples for FFHQ and 80 for DIV2K
>
> DIV2K Results
>
> | Method (DIV2K) | x8 PSNR | x8 LPIPS | x12 PSNR | x12 LPIPS |
> | --- | --- | --- | --- | --- |
> | Resample | 22.08 | 0.5480 | 20.60 | 0.6576 |
> | FlowDPS | 22.53 | 0.4837 | 21.47 | 0.5524 |
> | FlowDPS (8x ensemble) | 23.28 | 0.5157 | 22.06 | 0.5995 |
> | RSD | 23.45 | 0.5389 | 21.96 | 0.6836 |
> | FlowChef | 21.44 | 0.4898 | 20.20 | 0.5228 |
> | FlowChef (8x ensemble) | 22.60 | 0.5502 | 21.60 | 0.5931 |
> | Ours | 22.83 | 0.3627 | 21.05 | 0.4270 |
> | Ours (8x ensemble) | 23.79 | 0.4244 | 22.27 | 0.4930 |
>
> FFHQ Results
>
> | Method (FFHQ) | x8 PSNR | x8 LPIPS | x12 PSNR | x12 LPIPS |
> | --- | --- | --- | --- | --- |
> | Resample | 26.26 | 0.3860 | 25.31 | 0.4606 |
> | FlowDPS | 29.02 | 0.3659 | 27.74 | 0.4036 |
> | FlowDPS (8x ensemble) | 30.12 | 0.3267 | 28.65 | 0.3749 |
> | RSD | 29.69 | 0.3915 | 28.11 | 0.4624 |
> | FlowChef | 28.05 | 0.3303 | 26.56 | 0.3609 |
> | FlowChef (8x ensemble) | 29.54 | 0.3267 | 28.15 | 0.3602 |
> | Ours | 29.36 | 0.2028 | 27.42 | 0.2594 |
> | Ours (8x ensemble) | 30.94 | 0.2457 | 29.00 | 0.2999 |
>
> We therefore want the reviewers to be aware that our method is explicitly designed for posterior sampling with a generative prior, not for producing a single point estimate such as the MMSE solution. Consequently, **PSNR is not an appropriate metric for evaluating such generative approaches** (especially under heavily ill posed problems), as it penalizes diversity and favors averaging, which is contrary to the goal of realistic posterior sampling.
>
> **5. Real world non Gaussian noise**
>
> This goes beyond the scope of the paper. However, we want to emphasize that our method as well as other methods in the field are not limited to Gaussian noise. The data term can. also be adapted to other noise distributions (e.g. Poisson noise).
>
> **6. Training and Inference Time**
>
> We want to emphasize that our model is completely training free. We simply use the SD3 weights as they are. A runtime and memory consumption analysis for x12 super resolution can be seen below.
> Our hard data consistency can strongly influence the runtime and we also provide measurements with number of data term steps<=5 and a fast version using a "tinyVAE" of SD3.
>
> To validate that the usage of the tinyVAE or less steps does not degrades the performance noticeably we also provide a metrics for x12 Super Resolution on 100 samples of FFHQ:
>
> | Method | Runtime (s) ↓ | Memory (MB) ↓ | LPIPS ↓ | PSNR ↑ |
> | --- | --- | --- | --- | --- |
> | Resample | 88.02 | 19009.24 | 0.4606 | 25.31 |
> | FlowDPS | 34.15 | 12228.63 | 0.4036 | 27.74 |
> | RSD (no repulsion) | 21.19 | 12399.96 | 0.4624 | 28.11 |
> | FlowChef | 15.23 | 12227.72 | 0.3609 | 26.56 |
> | Ours (HDC, largeVAE) | 172.34 | 12389.37 | 0.259 | 27.42 |
> | Ours (HDC, tiny VAE) | 40.77 | 5960.22 | 0.256 | 27.59 |
> | Ours (5 data term steps, tiny VAE) | 22.46 | 5960.22 | 0.264 | 27.61 |
>
> **7. Novelty of Hard Data Consistency (HDC)**
>
> The decoupling of the data term is a novelty for variational posterior sampling approaches based on stochastic interpolants such as diffusion or flow matching models. A major drawback of most posterior sampling approach with stochastic interpolants is the need to approximate the likelihood term, because it is not tractable. Applying hard data consistency to the approximation does not necessarily bring an advantage and could event be harmful. However, in the variational formulation no approximation for the likelihood term has to be applied and therefore decoupling the data term and applying hard consistency can bring a large advantage as is shown in our ablations. We are the first to analyze and leverage this advantage.

---

> ### Comment · Reviewer_5yNE · 2025-08-05
>
> I have read the authors' rebuttals. The authors have addressed my concerns, including clarifying the innovation of this research relative to existing work and adding discussions on quantitative evaluation results and on non-Gaussian cases. Taking into account the innovation and contribution of this study, I have updated rating in the system.

---

### Official Review · Reviewer_Krp7 · 2025-07-02

**Clarity:** 2
**Significance:** 3
**Originality:** 3
**Rating:** 4
**Confidence:** 4

**Summary:**

The paper presents FLAIR (Flow-based Latent Adaptive Inference with deterministic Re-noising), a training-free variational framework that uses a flow-matching latent diffusion prior (Stable Diffusion 3) to solve inverse imaging tasks—single-image super-resolution, inpainting, and motion deblurring. Hard data-consistency (HDC) alternates regularizer updates with projections onto the measurement manifold, ensuring exact fidelity to the observation and Time-dependent calibrated weighting (CRW) scales the regularizer along the diffusion trajectory using offline error statistics

**Questions:**

1. Provide mean ± std over ≥ 3 random seeds and paired t-tests versus FlowDPS for LPIPS/FID.

2. Have you run a MOS or 2AFC study? If not, include at least NIQE or aesthetic-LPIPS to rule out metric over-fitting.

3. Does FLAIR handle non-Gaussian noise or compressive sensing (non-linear) operators? A small experiment would strengthen claims of “degradation-agnostic.”

4. How does CRW compare to a linear t weighting or a constant λ on all tasks?

**Ethical Concerns:**

["NO or VERY MINOR ethics concerns only"]

**Final Justification:**

Thank you for the follow-up clarifications and for completing the requested CRW ablation study across all tasks. The results provide helpful empirical evidence that CRW outperforms both linear and constant weighting strategies, and the inclusion of perceptual observations regarding inpainting quality is appreciated.

I will maintain my current score but acknowledge that this is a strong paper with clear technical contributions and well-documented experiments. I thank the authors for the constructive exchange throughout the rebuttal phase.

**Limitations:**

Authors acknowledge dependence on SD3 biases and new hyper-parameters but should also discuss: (i) high energy footprint of 50-step optimization on a GPU, and (ii) failure cases when the forward model is strongly non-linear.

**Paper Formatting Concerns:**

The checklist information that is supposed to be removed is not removed so it makes hard to follow the answers.

**Quality:**

3

**Strengths And Weaknesses:**

**Strengths:**
1. First integration of flow-matching priors and calibrated weighting into a unified inverse-problem solver.
2. This paper shows that a single latent flow prior can tackle multiple inverse problems without retraining.
3. Experiments on FFHQ and DIV2K show lower LPIPS/FID and sharper visuals than ReSample, FlowDPS, RSD and FlowChef across ×8/×12 SR, inpainting and deblurring.
4. Ablations confirm that DTA and CRW are crucial

**Weaknesses:**
1. DTA resembles classifier-free guidance; CRW is a heuristic weighting fitted offline hence incremental novelty.
2. Per-image solver needs 50 function evaluations on an RTX 4090—runtime not quantified.
3. Applicability limited to VAE-based latents of SD3; extension to pixel-space flows not shown.

---

> ### Author Rebuttal · Authors · 2025-07-27
>
> We thank the reviewer for acknowledging the novelty of our unified inverse-problem solver, the strong generalization of our flow prior across tasks, and the high quality of our results. We also appreciate your recognition of the importance of our ablations. Your feedback is much appreciated and has been incorporated into the manuscript.
>
> If our answers are satisfying, we kindly ask to consider raising your score.
>
> **1. DTA resembles classifier-free guidance; CRW is a heuristic weighting fitted offline hence incremental novelty**
>
> We want to thoroughly highlight, that the proposed deterministic trajectory adjustment does not share any resemblance to CFG. However, CFG can be used to increase the prompt following and therefore condition the model more strongly on the given text prompt. In the inverse problem setting we usually do not have good text prompts or no text prompts at all because the input is degraded and potentially we do not know what the original image depicts. With the deterministic trajectory adjustment we propose to use a "special noise vector" $\hat{x}_1$, which additionaly conditions the generative model on the current optimization estimate $\mu$. This ensures that the optimization does not jump between possible modes and instead stabilizes the regularization term.
> Other variational approaches like RED-diff or RSD also have to chose a weighting for the regularizer. We are the first to propose one which is motivated by the accuracy of the regularizer.
>
> **2. Runtime Concerns**
>
> This is not a new drawback of our method but one that all training free posterior sampling methods share. We decided to run all inverse problems using the same amount of neural functions evaluations. For certain problems the required number could be higher or lower. The amount of NFEs depends also on the generative model used.
>
> See a runtime and memory analysis below. Our hard data consistency can strongly influence the runtime and we also provide measurements with number of data term steps<=5 and a fast version using a "tinyVAE" of SD3.
>
> To validate that the usage of the tinyVAE or less steps does not degrades the performance noticeably we also provide a metrics for x12 Super Resolution on 100 samples of FFHQ:
>
> | Method | Runtime (s) ↓ | Memory (MB) ↓ | LPIPS ↓ | PSNR ↑ |
> | --- | --- | --- | --- | --- |
> | Resample | 88.02 | 19009.24 | 0.4606 | 25.31 |
> | FlowDPS | 34.15 | 12228.63 | 0.4036 | 27.74 |
> | RSD (no repulsion) | 21.19 | 12399.96 | 0.4624 | 28.11 |
> | FlowChef | 15.23 | 12227.72 | 0.3609 | 26.56 |
> | Ours (HDC, largeVAE) | 172.34 | 12389.37 | 0.259 | 27.42 |
> | Ours (HDC, tiny VAE) | 40.77 | 5960.22 | 0.256 | 27.59 |
> | Ours (5 data term steps, tiny VAE) | 22.46 | 5960.22 | 0.264 | 27.61 |
>
> **3. Pixel Space experiments**
>
> We thank the reviewer for the suggestion. Our focus was on high-resolution tasks and arbitrary scenes, where latent models such as SD3 offer clear benefits in terms of dataset scale and fidelity. Therefore, we did not include pixel-space experiments in the manuscript.
>
> However, we agree that such experiments are valuable for completeness, and we now provide them in the table below. s shown, our method outperforms previous works also in pixel space, demonstrating its broader applicability.
>
> | Method (Inpainting) | LPIPS ↓ | FID ↓ | SSIM ↑ | PSNR ↑ |
> | --- | --- | --- | --- | --- |
> | DDNM | 0.158 | 26.9 | 0.732 | 18.31 |
> | DPS | 0.195 | 30.2 | 0.689 | 20.49 |
> | Moment Matching | 0.161 | 28.8 | 0.728 | 20.59 |
> | $\Pi$GDM | 0.195 | 30.2 | 0.689 | 20.49 |
> | Ours | **0.097** | **14.2** | **0.831** | **21.87** |
>
> | Method (SR8) | LPIPS ↓ | FID ↓ | SSIM ↑ | PSNR ↑ |
> | --- | --- | --- | --- | --- |
> | DDNM | 0.199 | 31.9 | 0.635 | 23.59 |
> | DPS | 0.172 | 27.8 | 0.658 | 24.59 |
> | Moment Matching | 0.172 | 29.1 | 0.669 | 24.65 |
> | $\Pi$GDM | 0.157 | 26.5 | 0.677 | 24.98 |
> | Ours | **0.143** | **22.9** | **0.718** | **25.93** |
>
> **4. Statistical Relevance**
>
> We want to highlight that our method is training free and that variance in the reconstructed images is intentional. We test all methods with the same random seeds and evaluate over 1000 samples for FFHQ and 800 for DIV2K. pFID is evaluated on patches of 256x256 which results in 9000 samples and 7200 samples for evaluation. In table 5 in A.8 shows the means and standard deviations over different samples.
>
> For the rebuttal we also conducted experiments on 100 samples of FFHQ and 80 of DIV2K where we sampled 3 reconstructions per input for each method and computed the mean for each metric over all samples and the std of the means.
>
> | Method (FFHQ, SRx8) | LPIPS ↓ | FID ↓ | SSIM ↑ | PSNR ↑ |
> | --- | --- | --- | --- | --- |
> | FlowDPS | 0.3701 +- 0.0012 | 70.7 +- 1.45 | 0.7551 +- 0.0010 | 28.98 +- 0.008 |
> | RSD | 0.4678 +- 0.0001 | 102.9 +- 0.05 | 0.7362 +- 0.0001 | 28.45 +- 0.001 |
> | FlowChef | 0.3316 +- 0.0055 | 63.5 +- 1.45 | 0.7593 +- 0.0027 | 28.12 +- 0.072 |
> | Ours | 0.2039 +- 0.0048 | 40.5 +- 0.94 | 0.7970  +- 0.0228 | 29.74 +- 0.668 |
>
> | Method (FFHQ SRx12) | LPIPS ↓ | FID ↓ | SSIM ↑ | PSNR ↑ |
> | --- | --- | --- | --- | --- |
> | FlowDPS | 0.4073 +- 0.0002 | 77.6 +- 1.05 | 0.7391 +- 0.0006 | 27.71 +- 0.016 |
> | RSD | 0.5039 +- 0.0002 | 119.0 +- 0.12 | 0.7217 +- 0.0001 | 27.08 +- 0.001 |
> | FlowChef | 0.3626 +- 0.0050 | 81.1 +- 1.03 | 0.7283 +- 0.0027 | 26.62 +- 0.059 |
> | Ours | 0.2593  +- 0.0023 | 45.8 +- 1.51 | 0.7582 +- 0.0252 | 27.81 +- 0.660 |
>
> **5. User studies**
>
> A user study is the gold standard for image quality assessment, but was out of scope for this project. However, in a training free approach there are limited parameters to tune and metric overfitting is usually not an issue. Specifically, we conducted a small grid for FLAIR and all baselines methods to select optimal learning rates. All other parameters were kept fixed.
> Furthermore, LPIPS is already a distance metric which aligns very well with the human perception. The "realness" of the images on the other hand is usually evaluated with the FID metric.
> At this point we also want to emphasize that PSNR and SSIM are not ideal metrics to access the quality of the reconstructions, because they favor a MMSE prediction which is overly smooth and unrealistic. Consequently, any plausible texture shift, while perceptually correct, incurs a large $\ell_2$ penalty and reduces PSNR/SSIM.
>
> To highlight that our model can also be used to obtain good MMSE estimates, we also conducted ensemble predictions by running posterior sampling 8 times and averaging the result. The results show that ensembling increases PSNR values, but reduces LPIPS and confirms that our samples are indeed distributed around the posterior mean and that results very close to the posterior mean like the baseline methods are perceptually further away (LPIPS) from the ground truth compared to our samples.  We included the ensembling experiments in our supplementary material.
>
> 100 samples for FFHQ and 80 for DIV2K
>
> | Method (FFHQ) | x8 PSNR | x8 LPIPS | x12 LPIPS | x12LPIPS |
> | --- | --- | --- | --- | --- |
> | Resample | 26.26 | 0.3860 | 25.31 | 0.4606 |
> | FlowDPS | 29.02 | 0.3659 | 27.74 | 0.4036 |
> | FlowDPS (8x ensemble) | 30.12 | 0.3267 | 28.65 | 0.3749 |
> | RSD | 29.69 | 0.3915 | 28.11 | 0.4624 |
> | FlowChef | 28.05 | 0.3303 | 26.56 | 0.3609 |
> | FlowChef (8x ensemble) | 29.54 | 0.3267 | 28.15 | 0.3602 |
> | Ours | 29.36 | 0.2028 | 27.42 | 0.2594 |
> | Ours (8x ensemble) | 30.94 | 0.2457 | 29.00 | 0.2999 |
>
> | Method (DIV2K) | x8 PSNR | x8 LPIPS | x12 PSNR | x12 LPIPS |
> | --- | --- | --- | --- | --- |
> | Resample | 22.08 | 0.5480 | 20.60 | 0.6576 |
> | FlowDPS | 22.53 | 0.4837 | 21.47 | 0.5524 |
> | FlowDPS (8x ensemble) | 23.28 | 0.5157 | 22.06 | 0.5995 |
> | RSD | 23.45 | 0.5389 | 21.96 | 0.6836 |
> | FlowChef | 21.44 | 0.4898 | 20.20 | 0.5228 |
> | FlowChef (8x ensemble) | 22.60 | 0.5502 | 21.60 | 0.5931 |
> | Ours | 22.83 | 0.3627 | 21.05 | 0.4270 |
> | Ours (8x ensemble) | 23.79 | 0.4244 | 22.27 | 0.4930 |
>
> Furthermore a high PSNR does not mean that the results accurately follow the measurement. For x12 super resolution on DIV2K with FLAIR achieves 22.83 dB vs RSD with 23.45 dB. However, if we evaluate the PNSR in the measurement space to access how well the prediction follows the measurement y: PSNR(y, A($\hat{x}$)). FLAIR achieves 45.45dB in PSNR and RSD 40.28 dB. That means our prediction follows the measurement more closely even though the PSNR of the prediction compared to the ground truth is lower.
>
> **6. Non Gaussian noise and non-linear forward operators**
>
> The data term can also be adapted to non-Gaussian noise sources. E.g. Chung et al. Diffusion posterior sampling for general noisy inverse problems, ICLR 2023 elaborate on how to adapt the data term for Poisson distributed nose. We adapted the manuscript to clarify that our approach is not limited to only Gaussian noise models.
>
> Because SD3 is a latent model, the non-linear decoder of the SD3-VAE is part of our forward operator. Therefore, all tasks we evaluated on are non-linear. We also conducted experiments on non-linear motion deblurring (non-linearity through saturation effects and gamma correction) see tables below:
>
> Non-linear Motion Deblurring:
>
> | Method (FFHQ) | LPIPS ↓ | FID ↓ | SSIM ↑ | PSNR ↑ |
> | --- | --- | --- | --- | --- |
> | Resample | 0.446 | 118.1 | 0.792 | 25.21 |
> | FlowDPS | 0.413  | 83.0 | 0.738 | 27.27 |
> | RSD | 0.448 | 109.87 | 0.7441 | 27.27 |
> | FlowChef | 0.402 | 74.5 | 0.716 | 25.29 |
> | Ours  | 0.201 | 38.0 | 0.7851 | 29.37 |
>
> | Method (DIV2K) | LPIPS ↓ | FID ↓ | SSIM ↑ | PSNR ↑ |
> | --- | --- | --- | --- | --- |
> | Resample | 0.551 | 149.1 | 0.607 | 21.79 |
> | FlowDPS | 0.612 | 157.8 | 0.500 | 21.30 |
> | RSD | 0.626 | 177.3 | 0.542 | 21.81 |
> | FlowChef | 0.632 | 160.4 | 0.455 | 19.57 |
> | Ours  | 0.303 | 78.2 | 0.644 | 23.99 |
>
> **6. CRW compared to a linear $t$ weighting or a constant $\lambda$**
>
> Our CRW ablation is a comparison to $t$ weighting.

---

> > ### Comment · Reviewer_Krp7 · 2025-08-04
> >
> > The rebuttal effectively addressed most of my concerns, including statistical significance, runtime trade-offs, and applicability to pixel-space and non-Gaussian degradations. The clarification on deterministic trajectory adjustment (DTA) and variance-aware ensembling is appreciated.
> >
> > However, I remain unconvinced about the full justification of CRW. The authors claim to provide ablations versus linear and constant weighting strategies, yet the rebuttal does not include detailed comparative results across tasks. A clearer tabulated summary would strengthen the argument that CRW offers more than a heuristic tuning scheme.
> >
> > Given the high energy cost of inference and the potential overlap between DTA and existing guidance strategies, I maintain my score. Nonetheless, I acknowledge the paper's technical merit, careful engineering, and promising results across diverse inverse problems.

---

> > > ### Author Response · Authors · 2025-08-05
> > >
> > > We thank the reviewer for opening the discussion. As we pointed out in the rebuttal there is no resemblance between DTA and CFG. We also believe that there is no overlap between DTA and other guidance strategies. Could you point out which one you are referring to?
> > >
> > > Concerning the analysis of CRW, it is standard practice to present such ablations on a representative task, especially when the method is general-purpose and performance trends are consistent. Please note, that in our case, although the inverse problems differ in their forward operators, the calibrated regularizer weight is **not** computed for each task separately, but once for the prior used (SD3). Our method does not require more energy compared to the baselines. FLAIR uses the same amount of NFEs, but can take longer to run due to the hard data consistency (HDC). However, in the rebuttal we showed how the method can easily be speed up to match or exceed the speed of the baselines without compromising performance.
> > >
> > > We hope these clarifications help address your remaining concerns and would be grateful if you might consider them in your final evaluation. Please let us know if any further clarification would be helpful.

---

> > > > ### Author Response · Authors · 2025-08-06
> > > > **CRW Ablation Study**
> > > >
> > > > We completed the ablation study as requested for each of the analyzed tasks on the FFHQ dataset. CRW still outperforms the linear weighting (with t) and constant weighting. The differences on the inpainting task appear small, because we evaluate on the full image, but perceptually the differences especially on the inpainted areas are very large. Our method predicts crisper and more realistic looking images.
> > > >
> > > > The reason why the FID values are much higher in this ablation compared to our experiment table in the paper is because we only evaluated on 100 images for computational reasons. A smaller dataset size results in higher FID scores.
> > > >
> > > > | FFHQ            | SR8 - FID | SR8 - LPIPS | SR12 - FID | SR12 - LPIPS | Deblur - FID | Deblur - LPIPS | Inpainting - FID | Inpainting - LPIPS |
> > > > |-----------------|-----------|-------------|-------------|---------------|----------------|------------------|--------------------|----------------------|
> > > > | linear weight (t) | 64.7     | 0.317       | 71.7        | 0.361         | 60.2           | 0.276            | 32.6               | 0.152                |
> > > > | const. weight     | 54.2     | 0.292       | 79.4        | 0.389         | 51.1           | 0.2561           | 66.8               | 0.294                |
> > > > | CRW              | **40.3** | **0.203**   | **45.3**    | **0.260**     | **37.5**       | **0.225**        | **31.1**           | **0.147**            |

---

> > > > > ### Comment · Reviewer_Krp7 · 2025-08-07
> > > > >
> > > > > Thank you for the follow-up clarifications and for completing the requested CRW ablation study across all tasks. The results provide helpful empirical evidence that CRW outperforms both linear and constant weighting strategies, and the inclusion of perceptual observations regarding inpainting quality is appreciated.
> > > > >
> > > > > I will maintain my current score but acknowledge that this is a strong paper with clear technical contributions and well-documented experiments. I thank the authors for the constructive exchange throughout the rebuttal phase.

---

### Official Review · Reviewer_ybuv · 2025-07-03

**Clarity:** 3
**Significance:** 2
**Originality:** 2
**Rating:** 4
**Confidence:** 3

**Summary:**

This paper proposes FLAIR, which is a training-free variational framework for solving inverse problems using flow-based latent generative models. It replaces traditional score-based regularization with a flow-matching loss between the learned velocity field and a deterministic reparameterized trajectory.

**Questions:**

See Weakness.

**Ethical Concerns:**

["NO or VERY MINOR ethics concerns only"]

**Final Justification:**

The authors have addressed my concerns regarding [W1] the novelty of the work, [W2] the hyperparameter settings, and [W3] the evaluation of runtime and memory usage. I appreciate the clarifications and will maintain my current score as borderline accept.

**Limitations:**

yes

**Quality:**

3

**Strengths And Weaknesses:**

Strengths:
- The paper is well written, with a clear structure and smooth presentation.
- It introduces flow matching into a variational inference framework.
- The reconstructed images exhibit fine details with strong perceptual quality.
- The experimental section is thorough with comprehensive baseline comparisons and well-designed ablation studies.

Weaknesses:
- The core components, such as flow matching, variational inference, and hard data consistency, are all drawn from prior works. The paper mainly assembles these ideas into a cohesive system without introducing fundamentally new algorithms or theoretical insights.
- The method introduces several non-trivial hyperparameters, such as $\lambda_{\mathcal{R}}(t)$, $\alpha$, and gradient descent learning rate. A sufficient sensitivity analysis for different tasks would be helpful.
- Although the method is training-free, it involves repeated evaluation of learned velocity fields, gradient steps, and projection steps, which could be computationally expensive. A runtime or memory comparison with baseline methods would be helpful.
- L179 should be subsection A.4, not A.3. L222 should be subsection A.9, not A.5.

---

> ### Author Rebuttal · Authors · 2025-07-27
>
> We thank the reviewer for acknowledging the novelty of integrating flow matching into a variational framework, the high quality of our results, and the comprehensiveness of our experiments. Your feedback is much appreciated and we we incorporated it into our manuscript.
>
> If our answers are satisfying, we kindly ask to consider raising your score.
>
> **1. Core components are an assembly of known ideas**
>
> We would like to clarify that our work introduces several novel contributions beyond assembling existing techniques:
>
> - **Variational posterior sampling for flow matching:**
>
>     We are the first to introduce a variational approach for posterior sampling with flow matching models. The derivation of this method is non-trivial and is provided in detail in Appendix A.1–A.3.
>
> - **Deterministic trajectory adjustments:**
>
>     Our proposed deterministic trajectory adjustment is a crucial and completley novel component of the method, and our ablation studies show it yields the largest improvements in LPIPS. This adjustment is theoretically motivated and has not been previously explored in the literature.
>
> - **Model accuracy-weighted regularization (CRW):**
>
>     We are the first to propose using model accuracy as a weight for the regularization term (CRW), which is feasible only for variational posterior sampling approaches. This is not possible for non-variational methods.
>
> - **Hard data consistency in a variational framework:**
>
>     While hard data consistency has been used in other works, our framework (FLAIR) is, to our knowledge, the first to apply it within a variational setting.
>
>
> Our work is positioned within the field of *training-free posterior sampling using generative priors*. Given that flow matching models represent the current state-of-the-art in image generation, we believe it is both timely and important to develop posterior sampling methods that are specifically compatible with these powerful generative priors.
>
> **2. Non-trivial hyperparameters**
>
> The hyperparameters we introduce are $\lambda_{\mathcal{R}(t)}$ and $\alpha$, both of which are fixed across all our experiments and datasets. The gradient descent learning rate for the data term must be selected depending on the chosen task, which is also the case for all baseline methods.
>
> Regarding the CRW, we explore alternative weightings in Appendix A.6. Additionally, our ablation study without CRW corresponds to setting $\lambda_{\mathcal{R}(t)} = t$.
>
> We also note that, in variational approaches, a function for $\lambda_{\mathcal{R}(t)}$ must always be specified (e.g., RSD and RED-diff both use $\lambda_{\mathcal{R}(t)} = t$). Our contribution is to make this choice dependent on the model error.
>
> **3. Runtime Memory Analysys**
>
> Runtime optimization was not the scope of our project. Our baseline methods are evaluated with the same number of neural function evaluations. The methods may differ in the number of gradient steps taken on the data term, which depends on the early stopping constant chosen.
>
> Our method can also be deployed with a fixed number of data term steps (e.g., 5), if a deterministic runtime is required. If runtime or memory footprint are of importance one can switch to a "tinyVAE" version of SD3.
>
> To validate that the usage of the tinyVAE or less steps does not degrades the performance noticeably we also provide a metrics for x12 Super Resolution on 100 samples of FFHQ:
>
> | Method | Runtime (s) ↓ | Memory (MB) ↓ | LPIPS ↓ | PSNR ↑ |
> | --- | --- | --- | --- | --- |
> | Resample | 88.02 | 19009.24 | 0.4606 | 25.31 |
> | FlowDPS | 34.15 | 12228.63 | 0.4036 | 27.74 |
> | RSD (no repulsion) | 21.19 | 12399.96 | 0.4624 | 28.11 |
> | FlowChef | 15.23 | 12227.72 | 0.3609 | 26.56 |
> | Ours (HDC, largeVAE) | 172.34 | 12389.37 | 0.259 | 27.42 |
> | Ours (HDC, tiny VAE) | 40.77 | 5960.22 | 0.256 | 27.59 |
> | Ours (5 data term steps, tiny VAE) | 22.46 | 5960.22 | 0.264 | 27.61 |

---

> > ### Comment · Reviewer_ybuv · 2025-08-05
> >
> > Thank you for addressing my concerns regarding [W1] the novelty of the work, [W2] the hyperparameter settings, and [W3] the evaluation of runtime and memory usage. I appreciate the clarifications and will retain my current score of borderline accept.

---

### Note · Authors · 2025-08-12

We thank all reviewers for their constructive feedback and for recognizing FLAIR’s technical merit, strong perceptual performance, and comprehensive evaluation. We are glad that our clarifications during the rebuttal resolved most concerns and all four reviewers converged to a positive assessment of our work.

**Novelty & Contributions:**

FLAIR is the first *variational* posterior sampling framework tailored for **flow matching models**, enabling training-free, task-agnostic inverse problem solving. Our main innovations are:

1. **Variational posterior sampling for flow matching** (derivation in App. A.1–A.3).
2. **Deterministic Trajectory Adjustment (DTA)** — conditioning on both random and deterministic “special noise” vectors to stabilize optimization and recover diverse, atypical modes.
3. **Calibrated Regularizer Weight (CRW)** — weighting based on model accuracy, independent of the task, outperforming linear/constant schemes across SR, inpainting, and deblurring.
4. **Hard Data Consistency (HDC) in a variational setting** — decoupling data and regularization terms without likelihood approximations.

**Reviewer Concerns Addressed:**

- **Runtime/Memory:** We showed fast variants (tinyVAE, ≤5 or even 1 data-term step) with minimal quality loss, matching or exceeding baseline speeds.
- **CRW Justification:** Ablations across all tasks confirm consistent gains over linear/constant weighting.
- **Applicability:** Demonstrated on pixel-space flows, and non-linear forward models, extension to non-Gaussian noise are discussed in related literature
- **Metrics:** We clarified the perception–distortion tradeoff; FLAIR targets realistic posterior samples (low LPIPS/FID), not MMSE estimates (high PSNR). Ensembling recovers PSNR if desired.
- **Relation to prior work:** DTA differs fundamentally from CFG or PnP-flow-style re-noising by incorporating deterministic conditioning, guiding updates toward measurement-consistent modes.

**Impact:**

FLAIR achieves state-of-the-art perceptual quality and sample diversity across diverse degradations using a pretrained latent flow prior. It offers a principled, efficient, and general framework for future inverse-problem research with generative models.

We thank the reviewers for the engaging and constructive discussion, and we look forward to contributing FLAIR to the community.

---

### Decision · Program_Chairs · 2025-09-17

**Decision:**

Accept (poster)

**Comment:**

This paper presents FLAIR, a novel and well-motivated variational framework for solving inverse problems using powerful flow-based generative priors like Stable Diffusion 3. Through innovations like Deterministic Trajectory Adjustment and Calibrated Regularizer Weighting, the method achieves state-of-the-art results with excellent perceptual quality across a range of challenging tasks. After a comprehensive rebuttal that included new experiments addressing all initial concerns about novelty, runtime, and design choices, all four reviewers converged on a positive recommendation for acceptance.